



# Hydrogen peroxide in the upper tropical troposphere over the Atlantic Ocean and western Africa during the CAFE-Africa aircraft campaign

Zaneta Hamryszczak[1], Dirk Dienhart[1], Bettina Brendel[1], Roland Rohloff[1], Daniel Marno[1], Monica Martinez[1], Hartwig Harder[1], Andrea Pozzer[1,4], Birger Bohn[2], Martin Zöger[3], Jos Lelieveld[1,4], and Horst Fischer[1]

[1]Atmospheric Chemistry Department, Max Planck Institute for Chemistry, Mainz, 55128, Germany
[2]Institute of Energy and Climate Research, IEK-8: Troposphere, Forschungszentrum Jülich GmbH, Jülich, 52428, Germany
[3]Flight Experiments, German Aerospace Center (DLR), Oberpfaffenhofen, 82234 Germany
[4]Climate and Atmosphere Research Center, The Cyprus Institute, Nicosia, 1645, Cyprus

*Correspondence:* Zaneta Hamryszczak (z.hamryszczak@mpic.de) and Horst Fischer (horst.fischer@mpic.de)

**Abstract.** This study focuses on the distribution of hydrogen peroxide ($H_2O_2$) in the upper tropical troposphere at altitudes between 8 and 15 km based on *in situ* observations during the Chemistry of the Atmosphere – Field Experiment in Africa (CAFE-Africa) campaign conducted in August–September 2018 over the tropical Atlantic Ocean and western Africa. The measured hydrogen peroxide mixing ratios in the upper troposphere show a nearly uniform latitudinal distribution with locally increased levels (up to 1 $ppb_v$) within the Intertropical Convergence Zone (ITCZ), over the African coastal area, as well as during measurements performed in proximity of the tropical storm Florence (later developing into a hurricane), indicating the influence of convective transport processes. The measurements are compared to observation-based photostationary steady-state (PSS) calculations and numerical simulations by the global EMAC model. North of the ITCZ, PSS calculations produce lower $H_2O_2$ mixing ratios relative to the observations. Here observed mixing ratios exceed the PSS calculations by up to a factor of 2. On the other hand, PSS calculations overestimate the $H_2O_2$ mixing ratios south of the ITCZ by a factor of up to 3. The significant influence of convection in the ITCZ and the enhanced presence of clouds towards the southern hemisphere indicate contributions of atmospheric transport and cloud scavenging in the probed region. Differences between $H_2O_2$ observations and simulations of local PSS indicate that convective transport in the ITCZ region and consequent redistribution of $H_2O_2$ towards the north and south impacts the spatial distribution of $H_2O_2$ in the upper troposphere.

Simulations performed by EMAC analogously overestimate hydrogen peroxide levels particularly in the southern hemisphere, most likely due to underestimated cloud scavenging. Latitudinal distribution analysis indicates a gradient from the equator towards the subtropics both in the EMAC simulations and the PSS calculations. On the other hand, the measurements display nearly uniform mixing ratios of the species in the upper troposphere with a slight decrease from the ITCZ towards the subtropics, indicating a relatively low dependency on the solar radiation inclination and the corresponding photolytic activity. The highest deviations relative to the observations correspond with the underestimated hydrogen peroxide loss due to enhanced cloud presence, scavenging, and rainout in the ITCZ and towards the south.



## 1 Introduction

The key role of hydrogen peroxide in the oxidative chemistry of the troposphere is well acknowledged (Lelieveld and Crutzen, 1990; Crutzen et al., 1999). On the one hand, $H_2O_2$ serves as a reservoir of $HO_x$ (OH + $HO_2$ = $HO_x$) species, which are the most prominent oxidants controlling the self-cleansing capacity of the atmosphere (Levy, 1971; Logan et al., 1981; Kleinman, 1991). On the other hand, hydrogen peroxide can oxidize $SO_2$ and to a minor extent $NO_2$ and convert them into $H_2SO_4$ and $HNO_3$ in clouds, rain, and fog, leading to their acidification (Hoffmann and Edwards, 1975; Penkett et al., 1979; Robbin Martin and Damschen, 1981; Damschen and Martin, 1983; Calvert et al., 1985).

The most prominent pathway leading to $H_2O_2$ production is the recombination of $HO_2$ radicals (R4). These precursors of $H_2O_2$ are mainly produced via oxidation of carbon monoxide by OH radicals, which are formed during the photolysis of ozone and the subsequent reaction of $O^1D$ with water vapor in the troposphere (R1 – R3). The budget of $H_2O_2$ is thus controlled by the fate of the $HO_2$ radical, which is determined by a competition between its self-reaction leading to $H_2O_2$ formation and its conversion to OH via reaction with NO and $O_3$.

$$O_3 \xrightarrow{h\nu\,(\lambda<319\,nm)} O_2 + O(^1D) \tag{R1}$$

$$O(^1D) + H_2O \rightarrow 2\,OH \tag{R2}$$

$$OH + CO + O_2 \rightarrow HO_2 + CO_2 \tag{R3}$$

$$2\,HO_2 \rightarrow H_2O_2 + O_2 \tag{R4}$$

$$H_2O_2 \xrightarrow{h\nu\,(\lambda<360\,nm)} 2\,OH \tag{R5}$$

$$H_2O_2 + OH \rightarrow HO_2 + H_2O \tag{R6}$$

The global distribution of hydrogen peroxides is not only dependent on the chemical composition of the atmosphere but also on meteorological conditions. The amount of hydrogen peroxide is strongly dependent on the availability of water vapor and near-UV radiation (Jacob and Klockow, 1992; Perros, 1993; Slemr and Tremmel, 1994; Snow, 2003; Snow et al., 2007; Klippel et al., 2011). Towards the tropopause as well as towards the poles, the amount of water vapor generally decreases, resulting in a reduced primary production of $HO_x$ radicals. Additionally, with increasing altitudes near-UV radiation and therefore $H_2O_2$ photolysis increase leading to a pronounced production of OH via $H_2O_2$ photolysis in R5 (Jaeglé et al., 1997; Jaeglé et al., 2000; Faloona et al., 2000; Faloona et al., 2004; Lee et al., 2000). On the other hand, the availability of hydrogen peroxide precursors is diminished with increasing latitude, due to the decreasing inclination of solar radiation and reduced amount of water vapor towards the poles. Physical loss of hydrogen peroxide occurs through deposition processes promoted by its high solubility (Walcek, 1987; Chang et al., 2004; Nguyen et al., 2015). The aqueous uptake and subsequent removal of $H_2O_2$ strongly depends upon the uptake by aerosols and clouds (O'Sullivan et al., 1999). In effect, the vertical distribution of $H_2O_2$ often follows an inverted c-shape with decreased mixing ratios within the boundary layer and the upper troposphere and a local maximum in the middle troposphere at altitudes between 2 and 5 km. Additionally, observations indicate a decreasing trend approximately from the equator towards the north and south (Daum et al., 1990; Heikes, 1992; O'Sullivan et al., 1996; Weinstein-Lloyd et al., 1998; Snow, 2003; Snow et al., 2007; Klippel et al., 2011).





In general, the global budget of hydrogen peroxide is significantly affected by anthropogenic as well as natural emissions of nitrogen oxides. In urban areas, the formation of hydrogen peroxide is diminished by the increased mixing ratios of $NO_x$ ($NO_x = NO_2 + NO$) derived from anthropogenic sources, as the self-reaction of $HO_2$ to $H_2O_2$ is suppressed by the much faster reaction of $HO_2$ with NO (Lee et al., 2000; Reeves and Penkett, 2003). In contrast, biomass burning events lead to significant

injections of additional hydrogen peroxide through primary as well as secondary chemical production (Lee et al., 1997; Rinsland et al., 2007; Snow et al., 2007; Allen et al., 2022). Finally, convection processes are considered to increase the mixing ratios of hydrogen peroxide in the upper troposphere (Jaeglé et al., 1997; Jaeglé et al., 2000; Klippel et al., 2011; Bozem et al., 2017). Especially within the Intertropical Convergence Zone (ITCZ), convective processes play a key role in the transport of a large suite of trace species to higher tropospheric layers. The ITCZ is a low-pressure region, which marks the meeting zone

of airmasses transported from both hemispheres and constitutes the ascending branch of the Hadley circulation (Waliser and Gautier, 1993). Due to the high sea surface temperatures, strong solar radiation and increased air humidity, this band-like area near the equator is mainly characterized by highly dynamic weather phenomena, namely, large convective cumulonimbus clouds penetrating deep into the upper troposphere (Hastenrath and Lamb, 1977; Waliser and Gautier, 1993). Thus, convective transport processes might contribute to increased levels of hydrogen peroxide in the upper troposphere and promote elevated

$HO_x$ levels via subsequent photochemical processes involving $H_2O_2$ degradation (Jaeglé et al., 1997; Jaeglé et al., 2000). Numerous airborne measurements of hydrogen peroxide were performed over the past decades over the Atlantic Ocean and in proximity to the ITCZ. The majority of these studies focused on the troposphere in the northern hemisphere, providing an overview on the vertical and latitudinal distribution of hydrogen peroxide.

In September and October 1992, as part of the NASA's Global Tropospheric Experiment (GTE) program, the Transport and

Atmospheric Chemistry Near the Equatorial Atlantic (TRACE A) mission took place over the Atlantic Ocean. The mean observed mixing ratio of hydrogen peroxide was approximately 0.2 $ppb_v$ in the upper troposphere (8–12 km) (Prather and Jacob, 1997; O'Sullivan et al., 1999). During the Subsonic Assessment Ozone and Nitrogen Oxide Experiment (SONEX) campaign, which took place in autumn 1997 over the North Atlantic, mean values of 0.12 $ppb_v$ (median: 0.08 $ppb_v$) specifically in the upper troposphere were observed (Snow et al., 2007). Allen et al. (2013) presented satellite-based global distribution

data of $H_2O_2$ in the mid-to-upper troposphere obtained by the Atmospheric Chemistry Experiment (ACE) mission and reported mean levels of 0.10–0.28 $ppb_v$ in the upper tropical troposphere (>8 km), symmetrically decreasing towards the poles. During the Atmospheric Tomography Misson (ATom) performed in August 2016 (ATom-1) and October 2017 (ATom-3), mean values ranging between 0.05 $ppb_v$ up to 0.25 $ppb_v$ were measured over the Mid-Atlantic Ocean (-20°–20°N; Allen et al., 2022). Please note that the average values within the upper troposphere cited here are obtained from the information presented in the

figures in the study. Further, Hottmann et al. (2020) deduced mean (±1σ) and median hydrogen peroxide mixing ratios of 0.17 (±0.09) $ppb_v$ and 0.15 $ppb_v$, respectively, during the Oxidation Mechanism Observation (OMO) mission in summer 2015, which took place over the Arabian Peninsula, the eastern Mediterranean, and northern Indian Ocean covering the marine ITCZ region east of the African continent.



Numerous measurements have been performed in the marine tropical troposphere. In this study, we address the budget of
hydrogen peroxide specifically in the upper tropical troposphere within the equatorial Atlantic region with a main focus on the
ITCZ. Our objective was to study the distribution of trace gases and radicals over the Central Atlantic and the possible impact
of convection in the ITCZ on the abundance of $H_2O_2$ in the upper troposphere.

## 2 CAFE-Africa campaign

The distribution of hydrogen peroxide ($H_2O_2$) was measured in the free troposphere over the Atlantic Ocean during the
Chemistry of the Atmosphere: Field Experiment in Africa (CAFE-Africa) campaign. The major objective of the mission was
to investigate the large-scale distribution of trace gases, radicals, and aerosols in the tropical eastern Atlantic and along the
western coast of Africa. In particular, the influence of biomass burning emissions and long-distance pollution transport on the
atmosphere's oxidation capacity and the chemical processing of trace gases and aerosols in clean and polluted airmasses were
studied.
The campaign took place in August and September 2018 during the West African monsoon. During this period, 14
measurement flights were made over the Atlantic Ocean and the African coast with the High Altitude and LOng-Range
research aircraft (HALO) operating from the international airport on Sal, Cape Verde (16.75°N, -22.95°E). The flights focused
on the upper troposphere up to an altitude of 15 km with a few vertical profiles mostly in the northern hemisphere. The probed
area covered a latitudinal and longitudinal range from approximately -10°N–50°N and -50°E–15°E. An overview of the
corresponding flight dates and the objectives of the individual flights was presented by Tadic et al. (2021). The flight tracks
color-coded by GPS flight altitudes are presented in Fig. 1.





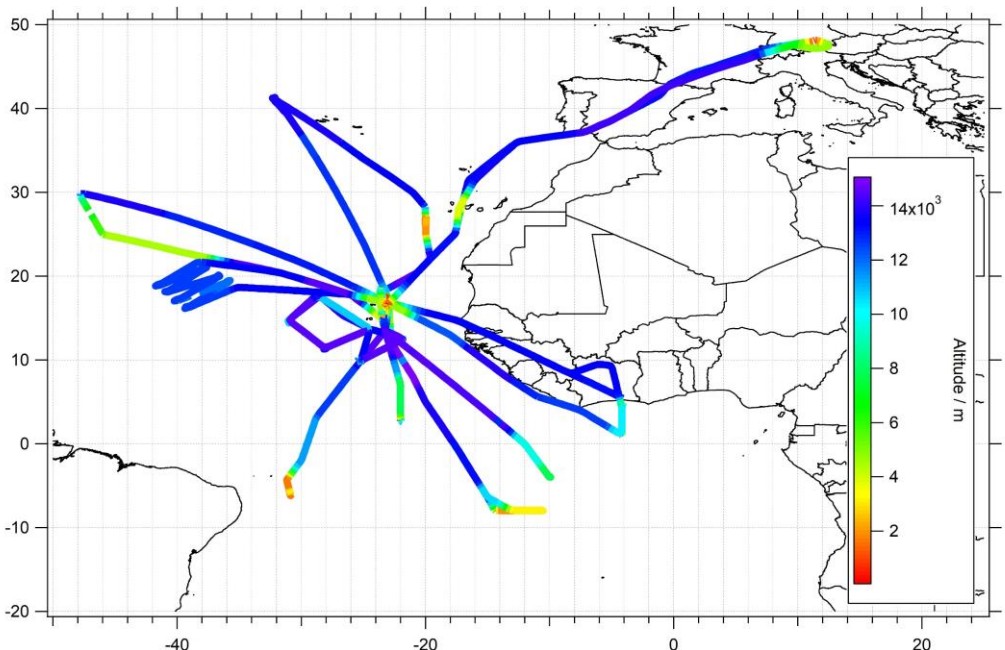

**Figure 1. Flight tracks and the sampled region during the CAFE-Africa campaign color-coded by the GPS flight altitude. The majority of flights were performed from the base of operations in Sal, Cape Verde.**

Related to the location of the base of operations on Cape Verde, the majority of the flights were performed in close proximity to the ITCZ, which allowed the study of tropical trace gases and aerosol distributions in both hemispheres. During the campaign, the ITCZ roughly covered positions between approximately 5°N and 15°N (Tadic et al., 2021). Information on the meteorological conditions with special emphasis on the total cloud coverage and convective precipitation located mostly within the ITCZ (5°N–20°N) are presented in the Supplement of this work (Figs. S1–3).

**3   Methods**

**3.1 Hydrogen peroxide measurements**

Hydroperoxides were measured as the sum of organic hydroperoxides and hydrogen peroxide and were determined using a wet chemical system named the Hydrogen Peroxide and Higher Organic Peroxides monitor (HYPHOP;(Klippel et al., 2011; Bozem et al., 2017; Hottmann et al., 2020; Hamryszczak et al., 2022) based on a previous design by Lazrus et al. (Lazrus et

al., 1985; Lazrus et al., 1986). The ambient air was sampled from the top of the aircraft fuselage via a trace gas inlet (TGI) with a ½" PFA liner that left the cabin again through a second exhaust line. From this bypass, a ¼" PFA sampling line is connected to a Teflon-coated membrane pump (Type MD 1C; Vacuubrand, Wertheim, Germany) and a pressure control unit regulating the pump speed to a line pressure of 1000 hPa (Constant Pressure Inlet; CPI). The CPI provides a constant inlet pressure covering external pressure variations between 1000 and 150 hPa. Following the CPI inlet, the ambient air passed



through a stripping coil with a buffered sampling solution (potassium hydrogen phthalate/NaOH; pH 6; stripping efficiency of 1 for hydrogen peroxide and between 0.6 and 1 for organic peroxides; Lee et al., 2000). The hydroperoxide solution was probed in two individual channels in response to addition of p-hydroxyphenyl acetic acid (POPHA) and horseradish peroxidase (HRP). The formed chemiluminescent 6,6'- dihydroxy-3,3'-biphenyldiacetic acid was detected via fluorescence spectroscopy using a Cd pen-ray lamp at 326 nm. The hydroperoxide-specific fluorescence (Guilbault et al., 1968) was detected at 400–420 nm

using photomultiplier tubes for each channel separately.

Hydrogen peroxide mixing ratios are then calculated from the difference between the entirety of the measured peroxides (Channel A) and the sum of organic ROOH hydroperoxides (Channel B), where $H_2O_2$ is selectively destroyed by the addition of catalase. Further information on the mixing ratios of individual organic peroxides cannot be provided by the monitor, due to the characteristics of the measurement technique. Please note that especially within the boundary layer and due to biomass

burning, a variety of organic peroxides might contribute to the total measured signal of organic peroxide (Fels and Junkermann, 1994; Slemr and Tremmel, 1994; Valverde-Canossa et al., 2005; Hua et al., 2008; Dienhart et al., 2022).

Prior to ambient measurements, both channels are simultaneously calibrated using a liquid standard (0.98 µmol $L^{-1}$) produced from serial dilution of a $H_2O_2$ stock solution. The $H_2O_2$ destruction efficiency in channel B corresponding to the added catalase was determined to be 0.95–0.98 based on liquid calibrations. In-flight background measurements were performed using

purified zero air, generated by a cartridge with silica gel (type IAC-502; Infiltec, Speyer, Germany) and hopcalite (type IAC-330; Infiltec, Speyer, Germany). Using a gas-phase calibration source (LDPE permeation devices), the $H_2O_2$ transmission efficiency through the inlet ($\pm$ $1\sigma$) was determined regularly by measuring the difference between the addition of the standard before and after the CPI and was found to be $0.61 \pm 0.06$. Due to a positive ozone interference, the $H_2O_2$ data was further corrected by subtraction of 0.056 ppb$_v$ $H_2O_2$/100 ppb$_v$ $O_3$ based on a scatter plot of hydrogen peroxide vs. ozone mixing ratios

in the lower stratosphere, assuming that ambient $H_2O_2$ above the tropopause is essentially zero. The total measurement uncertainty (TMU) of the monitor was estimated as:

$$TMU = \sqrt{((P)^2 + (US)^2 + (UOI)^2 + (UTE)^2)} \tag{1}$$

by considering the instrument's precision (P), uncertainty of the standard (US), uncertainty of the $H_2O_2$ transmission efficiency (UTE), and the uncertainty of the ozone interference (UOI). The determined precision with 1 sigma confidence interval was

determined from the reproducibility of the liquid calibrations performed during the campaign to be 1.3% at 5.46 ppb$_v$ for hydrogen peroxide and 0.8% at 5.64 ppb$_v$ for the organic hydroperoxides. The uncertainty of the standard was included in the instrument precision calculation. The uncertainty of the transmission efficiency was calculated to be 6%. The total measurement uncertainty was determined to be 9% for hydrogen peroxide and 41% for the sum of organic hydroperoxides. The total measurement uncertainty of organic hydroperoxides is increased by 40% due to the varying solubility of individual

organic hydroperoxides in aqueous solution, which ranges between 60% (e.g., MHP) and 100% (e.g., PAA). The instrumental time resolution was determined to be 122 s based on the calibration signal rise and fall time from 10% to 90% and 90% to 10%, respectively. The detection limit with a $2\sigma$ confidence was derived from the reproducibility of the in-flight background measurements as 15 ppt$_v$ for hydrogen peroxide and 6 ppt$_v$ for the sum of organic hydroperoxides, respectively. For the





purposes of this study, hydrogen peroxide data were filtered for stratospheric influences by removing all data points with ozone
mixing ratios higher than 100 ppb$_v$.

## 3.2 Measurement of other relevant species

GPS, temperature, pressure, and windspeed were obtained using the BAsic HALO Measurement And Sensor System,
BAHAMAS. Water vapor mixing ratios and the corresponding air humidity were measured with the Sophisticated Hygrometer
for Atmospheric ResearCh (SHARC) based on a tunable diode laser (TDL) setup (Krautstrunk and Giez, op. 2012). HO$_x$
radicals were measured by laser-induced fluorescence with the HydrOxyl Radical measurement Unit (HORUS; (Marno et al.,
2020). Spectrally resolved upward and downward actinic flux density was obtained with two spectroradiometers (Bohn and
Lohse, 2017). A brief overview of the campaign instrumentation, measurement methods, their TMU values, and the
corresponding technical references are listed in Table 1.

**Table 1. Overview of observed species with corresponding measurement method, total measurement uncertainty (TMU), and references regarding the instrumentation.**

| Measurement | Method | TMU | References |
|---|---|---|---|
| H$_2$O$_2$ | Chemiluminescence | H$_2$O$_2$: 9%; OrgPer: 41% | Hamryszczak et al., 2022 |
| HOx | Laser-induced Fluorescence (LIF; additional chem. conversion for HO$_2$) | 50% | Marno et al., 2020 |
| Actinic flux density | Spectroradiometer | 7–8% (15% for j(H$_2$O$_2$)) | Bohn and Lohse, 2017 |
| H$_2$O | TDLAS | 5% | Krautstrunk and Giez, 2012 |

## 3.3 Atmospheric Chemistry Model ECHAM/MESSy (EMAC)

For the purposes of this study, the *in situ* observations are compared to numerical simulations from the global chemistry and
climate 3-D model EMAC (ECHAM/MESSy for Atmospheric Chemistry, Jöckel et al., 2010). The model numerically
simulates the chemistry and dynamics of the troposphere and stratosphere using a large variety of submodels describing
chemical and meteorological processes and the influences arising from anthropogenic and natural emissions from continental
and marine environments (Jöckel et al., 2006). The basis of the atmospheric model is the 5$_{th}$ generation of the European Centre
HAMburg general circulation model (ECHAM5; Roeckner et al., 2003; Roeckner et al., 2006). The communication between
the various submodels is achieved by the Modular Earth Submodel System (MESSy; Jöckel et al., 2005; Jöckel et al., 2010;
Jöckel et al., 2016). Atmospheric chemistry is simulated by the Module for Efficiently Calculating the Chemistry of the
Atmosphere (MECCA) submodel (Sander et al., 2005; Sander et al., 2011; Sander et al., 2019), using the Mainz Organic
Mechanism (MOM) and photolysis rate calculations from a radiation transfer model (Sander et al., 2014; Sander et al., 2019).
Primary emissions and dry deposition as well as aqueous phase chemistry in clouds and cloud scavenging are simulated by the
ONLEM, OFFLEM, TNUDGE, and DRYDEP submodels (Kerkweg et al., 2006a; Kerkweg et al., 2006b) and the Scavenging
of Tracers submodel (SCAV; Tost et al., 2006). Anthropogenic emissions are based on the EDGARv4.3.2 inventory (European





Joint Center; JRC; Crippa et al., 2018) and are distributed vertically according to (Pozzer et al., 2009). Biomass burning emissions were simulated based on the Global Fire Assimilation System (GFAS; Kaiser et al., 2012). The model has a vertical and horizontal resolution of 47 vertical levels up to 0.01 hPa at T63 (i.e., approximately 1.8° x 1.8°), respectively, and a time resolution of 6 min. The model was further weakly nudged towards the ECMWF ERA-Interim data (Tadic et al., 2021). This model set-up has been extensively evaluated for different trace gases and aerosols (Pozzer et al., 2022). For comparison to

observations, the simulation results were interpolated on the GPS flight tracks using the S4D submodel (Jöckel et al., 2010).

### 3.4 Simulations based on photostationary steady-state conditions

The hydrogen peroxide mixing ratios in the upper troposphere under the assumption of photostationary steady-state conditions were calculated based on measured precursors and photochemical loss processes of hydrogen peroxide ($HO_2$, OH, $j(H_2O_2)$) and the rate coefficient data from Atkinson et al., 2004).

In the upper troposphere, the production rate $P(H_2O_2)$ of hydrogen peroxide due to the self-reaction of $HO_2$ can be calculated from Eq. (2). The photochemical loss rate of hydrogen peroxide, $L(H_2O_2)$ can be derived from $H_2O_2$ photolysis and the reaction with OH as shown in Eq. (3). Please note that because of low mixing ratios of other trace gases producing hydrogen peroxide in the upper troposphere, other potential photochemical sources and sinks of $H_2O_2$ were not considered. The equations are derived from the most prominent reaction pathways presented in Sect. 1 (R4–R6).

$$P(H_2O_2) = k_{HO_2+HO_2} \cdot [HO_2]^2 \tag{2}$$

$$L(H_2O_2) = \left( k_{H_2O_2+OH} \cdot [OH] + j(H_2O_2) \right) \cdot [H_2O_2] \tag{3}$$

The calculations of the rate coefficients were based on the measured parameters along the flight tracks according to Atkinson et al., 2004; Eqs. 4–5). Due to the water dependence of the hydrogen peroxide production rate coefficient, both expressions in Eq. (4) were further extended by the factor $1 + 1.4 \times 10^{-21}$ cm$^3$ $[H_2O]$ exp(2200/T).

$$k_{HO_2+HO_2} = 2.2 \cdot 10^{-13} \cdot \exp^{\frac{600}{T}} \text{cm}^3\text{s}^{-1} + 1.9 \cdot 10^{-33}[N_2] \cdot \exp^{\frac{980}{T}} \text{cm}^6\text{s}^{-1} \tag{4}$$

$$k_{H_2O_2+OH} = 2.9 \cdot 10^{-12} \cdot \exp^{\frac{-160}{T}} \text{cm}^3\text{s}^{-1} \tag{5}$$

Neglecting deposition and transport processes impacting the hydrogen peroxide budget, the mixing ratio of $H_2O_2$ was calculated via Eq. (6).

$$[H_2O_2]^{PSS} = \frac{[HO_2]^2 \cdot k_{HO_2+HO_2}}{[OH] \cdot k_{H_2O_2+OH} + j(H_2O_2)} \tag{6}$$

### 3.5 Data processing details

For the purpose of the present study, we used spatially interpolated EMAC simulations of $H_2O_2$, OH, $HO_2$, water vapor, $j(H_2O_2)$, temperature, and pressure to supplement the measurement observations. To synchronize the time stamp of the





simulated data with the measurement output, we calculated a mean of the measurement data with a matching temporal resolution of 6 min.

Vertical profiles of all species under investigation were calculated as 1000 m bins (means and medians) over the entire probed atmospheric column. Profile information is restricted to 30 take-offs and landings at Sal, while other areas are not considered due to a lack of statistically significant data.

The spatially resolved data based on measurements, PSS model calculations, and EMAC simulations were binned into 1° x 1° subsets over the full extension of the flight tracks in the upper troposphere (≥ 8 km).

The latitudinal distribution of the species was examined using 2.5° bins over the upper tropospheric region. Please note that due to a reduced amount of data in the lower troposphere, the analysis of spatial and latitudinal distributions was restricted to measurements performed in the upper troposphere.

## 4    Results

### 4.1  Observations of hydrogen peroxide during CAFE-Africa and previous airborne measurements

The observed mixing ratios of hydrogen peroxide during the CAFE-Africa campaign are presented as a latitude vs. longitude plot with mean mixing ratio values binned into a subset of 1° x 1° bins for the entirety of the upper troposphere (≥ 8 km) along the flight tracks (Fig. 2). The color scale represents the measured mixing ratio of $H_2O_2$.

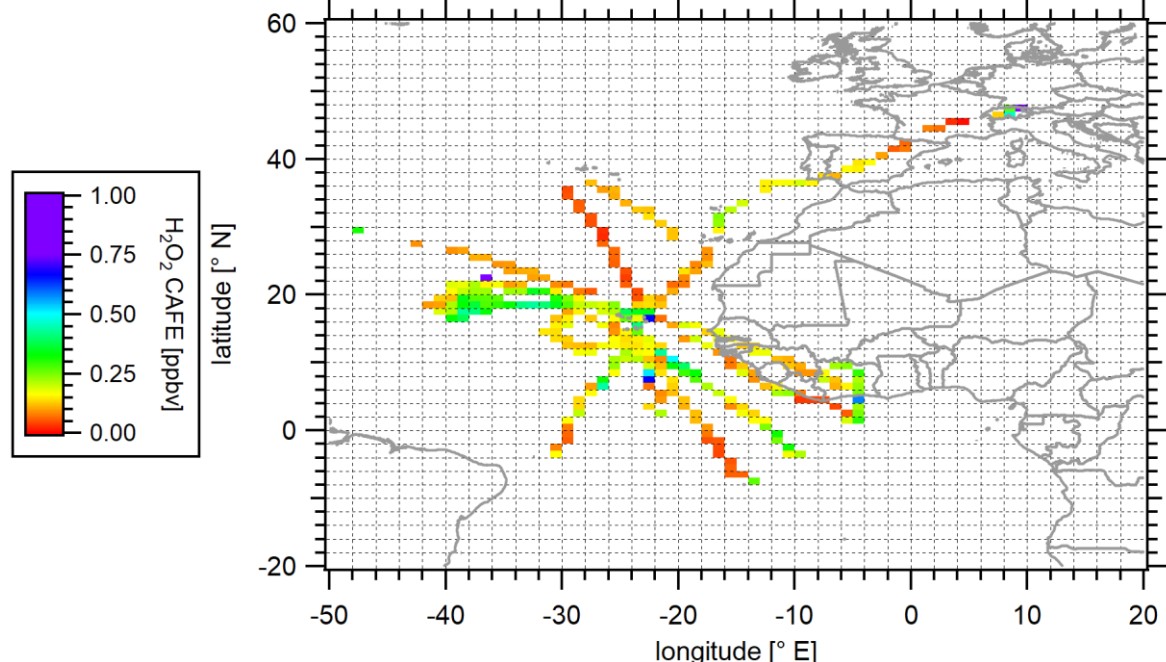

**Figure 2. Spatial distribution of measured hydrogen peroxide in the upper troposphere (≥8 km) during the CAFE-Africa campaign.**
**Data were binned into 1° x 1° bins over the full extension of the flight tracks.**




The mean ($\pm 1\sigma$) and median mixing ratios during the campaign were 0.18 ($\pm$ 0.13) ppb$_v$ and 0.15 ppb$_v$, respectively, with maximum hydrogen peroxide mixing ratios reaching 1.03 ppb$_v$. Slightly higher $H_2O_2$ levels were observed in the ITCZ (approx. 5°N–20°N), where locally mixing ratios up to 0.67 ppb$_v$ were observed. These maxima are most likely due to atmospheric transport of $H_2O_2$ in continental air masses transported from central and southern Africa into the upper troposphere. This is in

accordance with previous reports about increased hydrogen peroxide mixing ratios due to biomass burning and convective activity, elevating the $H_2O_2$ mixing ratios in the upper troposphere (Lee et al., 1998; O'Sullivan et al., 1999; Allen et al., 2022). Locally enhanced $H_2O_2$ was further observed during the measurement flight in close proximity to the tropical storm Florence on 2$^{nd}$ of September 2018 (approx. 18°N, - 38.5°E). Here, the mean mixing ratios were between 0.10 ppb$_v$ and 0.43 ppb$_v$ with a maximum of up to 0.94 ppb$_v$. Based on the high convective activity during the tropical storm, as reported by Nussbaumer et

al. (2021), the $H_2O_2$ mixing ratios were expected to rise due to the rapid transport of air masses from the MBL into the UT. Table 2 gives an overview of the estimated mean and median hydrogen peroxide mixing ratios measured during CAFE-Africa in relation to previous airborne measurements performed at a comparable latitudinal range.

**Table 2. Comparison of hydrogen peroxide mean and median mixing ratios (ppb$_v$) in the upper troposphere during CAFE-Africa with measurements from previous campaigns (TRACE A, SONEX, OMO, ATom-1 and ATom-3; O'Sullivan et al., 1999; Snow et**
**al., 2007; Hottmann et al., 2020; Allen et al., 2022).**

| | CAFE-Africa | TRACE A | SONEX | OMO | ATom-1 | | ATom-3 | |
|---|---|---|---|---|---|---|---|---|
| | 10°S–40°N | 40°S–15°N | 15°–60°N | 0–50°N | 20°–60°N | 20°S–20°N | 20°–60°N | 20°S–20°N |
| **Mean** | **0.18** | <0.20 | 0.12 | 0.16 | 0.55 | 0.61 | 0.18 | 0.40 |
| **Median** | **0.15** | 0.15 | 0.08 | 0.15 | 0.29 | 0.27 | 0.12 | 0.20 |

The mean and median values during CAFE-Africa are comparable to previously reported mixing ratios during TRACE A and OMO campaigns, which covered a comparable latitude and altitude range (Tab. 2). Enhanced mixing ratios for CAFE-Africa relative to observations during the SONEX campaign are most likely due to differences in the examined regional range, since
the later campaign focused on the north Atlantic. Mean and median values in the northern hemisphere (20–40°N; Tab. S1) during CAFE-Africa (0.14 $\pm$ 0.11 ppb$_v$ and 0.12 ppb$_v$, respectively) are comparable to observations during SONEX. During the ATom campaigns, slightly higher mean values were observed, although median values are comparable. This could be due to differences in the probed altitudes, since ATom measurements were generally restricted to altitudes below 12 km.

Based on the comparison with previous studies, the observed mixing ratios of hydrogen peroxide during CAFE-Africa fit well
into the general range of recent studies over the equatorial and subtropical Atlantic. The observed $H_2O_2$ distribution confirms further that mixing ratios of hydrogen peroxide in the upper troposphere seem to be far less dependent on latitude than those at lower altitudes. The latitudinal distribution of $H_2O_2$ during the CAFE-Africa displays a rather small symmetrical latitudinal decrease from the inner tropics into the subtropics.





**4.2 Comparison of measured hydrogen peroxide with photostationary steady-state and EMAC calculations**

In order to investigate the impact of deep convection in the ITCZ on the $H_2O_2$ budget in the upper troposphere, a comparison of the *in situ* data with the output of photostationary steady-state (achieved) calculations and EMAC simulations was performed. The complementary spatial distributions of the hydrogen peroxide levels were expressed as latitude versus longitude plots of mean mixing ratios aggregated over a spatial grid of 1° x 1° in the upper troposphere (≥ 8 km) (Fig. S4 in the Supplement).

The calculated PSS-$H_2O_2$ levels range between approximately 0.01 $ppb_v$ and 0.88 $ppb_v$ with mean (± 1σ) and median mixing ratios of 0.14 (± 0.16) $ppb_v$ and 0.07 $ppb_v$, respectively, which is a factor of 1.3 lower than the observations. PSS-based $H_2O_2$ mixing ratios tend to be higher at the southernmost coastal area (-2.5°N, -10.5°E), where the levels range between 0.40 and 0.88 $ppb_v$, and in the proximity to the tropical storm Florence at up to 0.40 $ppb_v$ (approx. 18°N, -38.5°E). Hydrogen peroxide mixing ratios simulated by EMAC vary between 0.10 $ppb_v$ and 0.75 $ppb_v$. The mean (±1σ) and median simulated mixing ratios are 0.30 (±0.19) $ppb_v$ and 0.29 $ppb_v$, respectively, with maximum mixing ratios up to 1.04 $ppb_v$ (-3.5°N, -9.5°E; Fig. S4b), which is slightly higher than the observations. The spatial distributions of the point-by-point ratio between PSS calculations and EMAC simulations versus the observations in the upper troposphere above 8 km are presented in Fig. 3, which give an overview on the local differences relative to the observations varying from low ratios (yellow) to high (deep blue) values. Please note that for resolution purposes the color scaling is restricted to ratios up to 4.5.

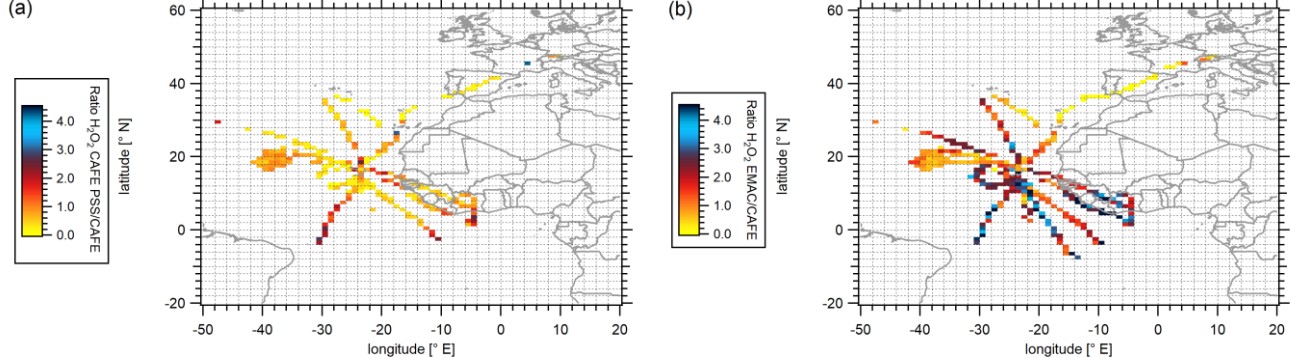

**Figure 3. Spatial distribution of $H_2O_2$(PSS)/$H_2O_2$(measurements) ratios (a) and $H_2O_2$(EMAC)/$H_2O_2$(measurements) ratios (b) in the upper troposphere (≥8 km) during the CAFE-Africa campaign. Data were binned into 1° x 1° bins over the full extension of the flight tracks.**

Generally, the $H_2O_2$(EMAC)/$H_2O_2$(measurement) ratios indicate a good agreement between the simulations and the measurements in the northern hemisphere (≥ 20°N; Fig. 3b), with values between 0.95 and 1.3. With decreasing latitudes, the model tends to overestimate hydrogen peroxide, with $H_2O_2$(EMAC)/$H_2O_2$(measurement) ratios increasing with latitude from approximately 2 to 4. Locally, most likely due to underestimated cloud scavenging, EMAC simulates highly elevated hydrogen peroxide with a factor of up to 14 higher than the measurements (4.5°N, -9.5°E).

Good agreement between the observation-based PSS calculations and the measurements was found in proximity to the tropical storm Florence. Here, the $H_2O_2$(PSS)/$H_2O_2$(measurement) ratios agree between 0.83 to 1.04. Beyond the measurements taken





here, the agreement with respect to the measured hydrogen peroxide levels is generally less satisfactory. Towards the southern subtropics as well as locally in the coastal area and at the base of operations in Sal (Cape Verde), the ratios increase to 2.8, indicating an overestimation of the hydroperoxide levels relative to the observations, similar to the EMAC simulations. On the

other hand, the PSS calculations tend to strongly underestimate hydrogen peroxide concentrations in the ITCZ (5–20°N) and in the northern part of the investigated region (≥ 20°N) by factors of up to 10 and 12, respectively. This indicates that, in the upper troposphere, the local photostationary steady-state conditions based on observed radical levels do not account for all sources and sinks of hydrogen peroxide or additional injections of the $H_2O_2$, which are most likely related to transport and cloud scavenging. The calculated difference between production (Eq. 2) and loss (Eq. 3) of hydrogen peroxide, (P–L)$H_2O_2$

based on the observations is expressed as a latitude vs. longitude plot with mean mixing ratios binned into a subset of 1° x 1° bins for the entirety of the upper troposphere (≥ 8 km) along the flight tracks (Fig. 4).

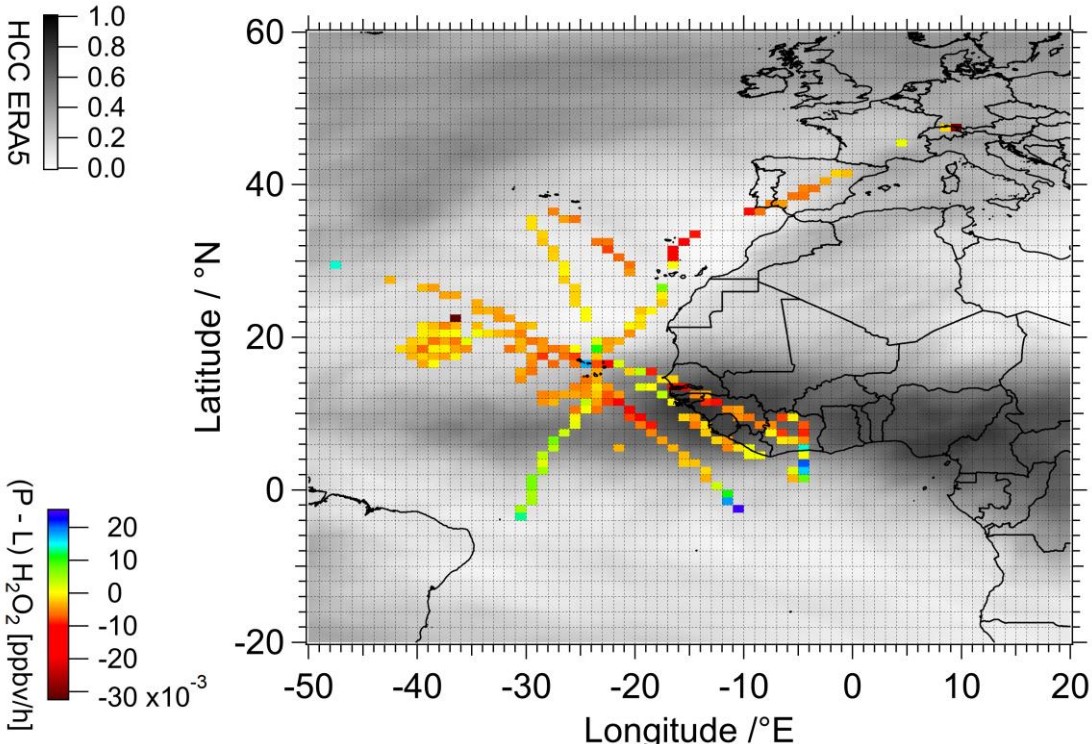

**Figure 4. Color-coded spatial resolution of calculated deviation from PSS based on the difference between the observed hydrogen peroxide photochemical production and loss. The calculated data were binned into 1° x 1° bins over the full extension of the flight**
**tracks. Shaded background is the average high cloud coverage (≥6 km) during the measurement period based on ERA5 reanalysis results (HCC; Hersbach et al., 2019).**

Generally, the majority of the probed region is loss-dominated, especially in the ITCZ (approx. 5°N–20°N) and towards the north, where an $H_2O_2$ deficit of up to approximately -0.01 ppb$_v$ h$^{-1}$ was determined. $H_2O_2$ production-dominated regions of up to 0.03 ppb$_v$ h$^{-1}$ are observed towards the south and in the coastal area. The difference between the photochemical production

and loss paths is directly linked to the deviations from the photostationary steady state and thus to the additional sources and



sinks of hydrogen peroxide on a local scale. These are associated with the local meteorological conditions and transport processes. Based on ERA5 reanalysis results, especially towards the south and in the coastal area, enhanced presence of clouds at altitudes above 6 km (gray shading in Fig. 4) and convective precipitation was observed during the measurement period (Hersbach, H. et al., 2019). At a mean (±1σ) horizontal windspeed of 14.3 (±7.3) m/s measured along the flight tracks and a
species lifetime of 3–4 days, transport towards the subtropics impacts $H_2O_2$ levels and further justifies the difference between the local PSS calculations and the observations.

Based on the coincidence with the latitudinal range of the ITCZ, the enhanced loss in the $H_2O_2$ budget relative to the PSS in the upper troposphere is most likely due to convective injection of $H_2O_2$ from lower layers into the upper troposphere and the subsequent redistribution of the species towards the north and south. Below, we show that, based on the comparison with
EMAC simulation output, convective transport is important for the budget of hydrogen peroxide in the upper troposphere not only in the ITCZ but also in the subtropics.

As discussed above, the comparison between observations and both PSS calculations and EMAC simulations indicate large deviations at the most southern latitudes that were visited by HALO. This is clearly demonstrated in Fig. 5, which shows observations, PSS calculations, and EMAC simulations of $H_2O_2$ as functions of latitude in the upper troposphere (above 8 km).
The mean values of each dataset with 6 min time resolution are binned into subsets of 2.5° of latitude for the investigated region from 6°S to 40°N. The lines and the complementary shading represent mean values and the supplementary standard deviations. The area of the ITCZ between 5°N and 20°N is highlighted by gray shading.

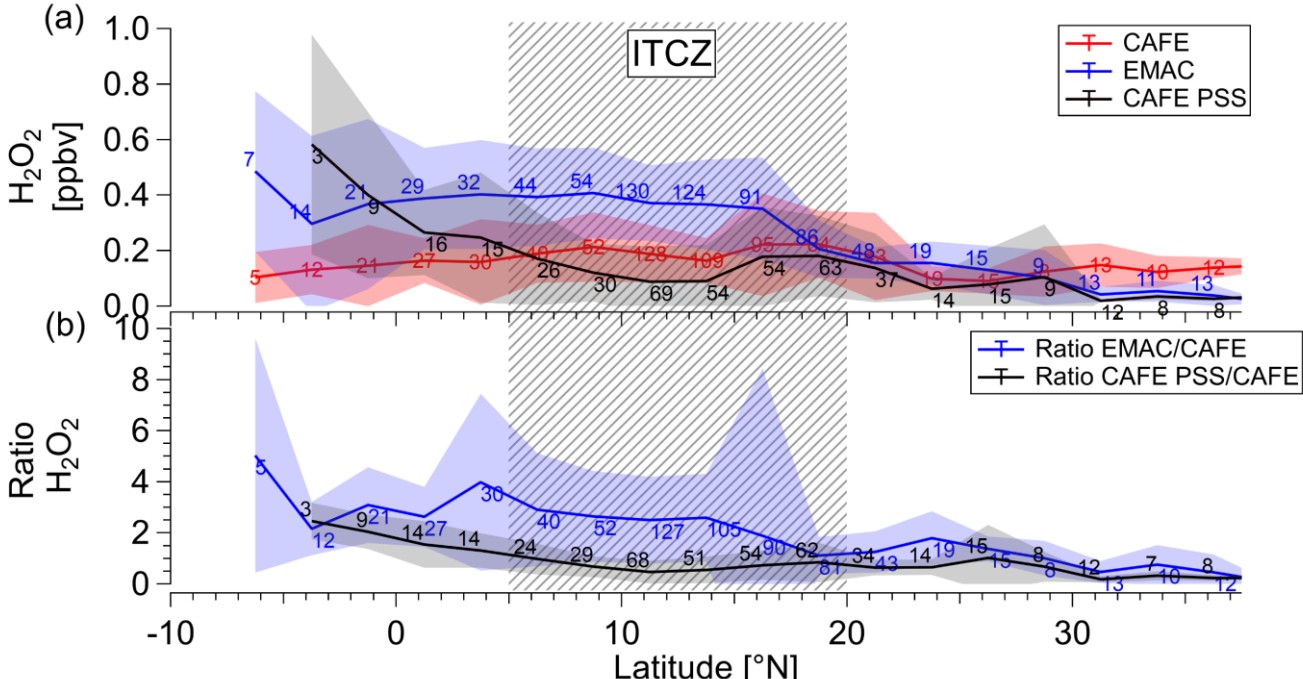

**Figure 5.** Latitudinal dependence of hydrogen peroxide mixing ratios (mean ± 1σ) compared to EMAC simulations and calculations
**based on PSS (red: CEFE-Africa; black: PSS CAFE-Africa; blue: EMAC; a) and calculated ratios between the simulations and the**





**observations and PSS-modelled calculations, and the observations, respectively (b). The data with 6 min time resolution and mean values was binned for 2.5° of latitude for altitudes ≥ 8 km. The corresponding numbers indicate the total amount of data points per bin. The shaded pattern marks the ITCZ region.**

Contrary to the calculations based on local photostationary steady-state conditions and the simulations by EMAC, the observations show very little latitudinal variation with mean values of approximately 0.1–0.2 ppb$_v$. A tendency towards slightly enhanced values is observed in the ITCZ, where the mixing ratios increase up to 0.22 ppb$_v$. However, considering the standard deviation range (up to 0.19 ppb$_v$), the rather flat distribution in the upper troposphere indicates nearly constant hydrogen peroxide levels throughout the whole investigated tropospheric region (see also Tab. S1).

Both the EMAC-simulated and PSS-calculated latitudinal hydrogen peroxide distributions display decreasing trends with increasing latitudes towards the north. The highest mean values of 0.49 (± 0.29) ppb$_v$ for PSS and 0.583 (± 0.40) ppb$_v$ for EMAC are found in the southernmost part of the probed region. The elevated levels of $H_2O_2$ in the EMAC simulations starts already in the ITCZ, while the PSS calculations only increase south of 5°N.

Overall $H_2O_2$ mixing ratios from the PSS calculations decrease from the equator towards the subtropics with at a rate of -0.004 (±0.002) ppb$_v$ per degree of latitude. At approximately 5°N, the PSS-based mixing ratios decrease to a range of 0.09 ppb$_v$ to 0.25 ppb$_v$ and tend to fall below the measured levels at the northern edge of the examined region by a factor of up to 5 (at approx. 32°N). The mixing ratios of hydrogen peroxide simulated by EMAC remain elevated at approximately 0.30 ppb$_v$ to 0.35 ppb$_v$ from 6°S to 15°N, yielding $H_2O_2$(EMAC)/$H_2O_2$(measurements) ratios of 2.2 to 2.5. Further north of 15°N, the $H_2O_2$ levels in EMAC decrease by almost half to 0.15 ppb$_v$, resulting in $H_2O_2$(EMAC)/$H_2O_2$(measurements) ratios between 1.1 and 1.2 and a rather good agreement between simulations and observations. The linear fit applied to the latitudinal distribution of simulated hydrogen peroxide levels displays a mean (±1σ) decrease of -0.008 (±0.001) ppb$_v$ per degree of latitude. An overview of numerical values for measured means (±1σ) and medians, PSS-calculated and EMAC-simulated hydrogen peroxide levels subdivided into three hemispheric regions, the northern (approximately 20°N–40°N), ITCZ (approximately 5°N < 20°N), and southern hemisphere (approx.-10°N < 5°N) is given in Tab. S1 in the Supplement of this work. Please note that the steady-state calculations, which are based on observed levels of OH, HO$_2$, and $H_2O_2$ photolysis rates, only account for photochemical production and loss of hydrogen peroxide. EMAC simulations additionally account for vertical and horizontal transport processes, as well as local losses due to cloud scavenging (Hamryszczak et al., 2022). Thus, deviations between local photostationary steady-state budget calculations and EMAC simulations can be used to identify and quantify the impact of convective processes in the ITCZ on the budget of $H_2O_2$ in the upper troposphere. This requires that the EMAC model correctly simulates precursors (HO$_2$) and sinks (OH, $H_2O_2$ photolysis) of photochemical $H_2O_2$ formation.

A comparison between observations and EMAC simulations for the basic species that affect photochemical production (HO$_2$) and loss (OH and $H_2O_2$ photolysis rates) of $H_2O_2$ reveals that while HO$_2$ is adequately reproduced by EMAC, the simulations tend to underestimate OH and the photolysis frequencies, in particular, south of 15°N, where EMAC underestimates OH by up to 0.500 ppt$_v$ (Fig. S5b). At the same time, model results partly tend to overestimate HO$_2$ by approximately 3.5 ppt$_v$, indicating issues with HO$_2$/OH partitioning. Since the production of OH in the UT depends to an extent on the reaction of HO$_2$





with NO, these might be associated with the underestimation of NO by EMAC in the southern hemisphere as shown by Tadic et al. (2021). Additionally, the measured $H_2O_2$ photolysis frequencies show minor discrepancies with those from the model (Fig. S5c). Due to the overestimated $HO_2$ mixing ratios, EMAC simulates higher levels of $H_2O_2$. At the same time, the lower OH mixing ratios and the underestimated $H_2O_2$ photolysis frequencies might cause decreased loss rates, thus leading to underestimation of the $H_2O_2$ loss paths. Therefore, overestimation of the photochemical source and underestimation of the

photochemical $H_2O_2$ sinks by EMAC explain the differences between PSS calculations and simulations at southern latitudes; however, these observations fail to explain the observed differences between PSS calculations and EMAC simulations versus the observations south of the ITCZ. Also, it is highly unlikely that a measurement error is responsible for the discrepancies, as this would have to be local and restricted to the most southerly latitudes. Potential causes leading to the observed discrepancy between the measurement and the simulations might be an underestimation of cloud scavenging and removal of hydrogen

peroxide by wet and dry deposition in the ITCZ and also south of this region. A number of flights south of 15°N were performed in close proximity to the western shores of Africa or even over land (Fig. 2), close to enhanced convective precipitation (Fig. S3). Based on the ERA5 reanalysis results on cloud coverage during the measurement period (Hersbach, H.et al., 2019; Fig. S2), we hypothesize that the air masses probed in this area were affected by cloud processing especially in the UT, causing the deviations towards the model calculations.

Since the amount of the $H_2O_2$ in the upper atmosphere depends on convective transport from the lower troposphere as well as on losses caused by cloud droplets and the subsequent rainout, it is important that EMAC simulations and PSS calculations reproduce the levels of hydrogen peroxide in the middle troposphere and the boundary layer. In Fig. 6, averages of hydrogen peroxide mixing ratios based on *in situ* observations, PSS calculations, and EMAC simulations and the corresponding $H_2O_2(EMAC)/H_2O_2(measurements)$ and $H_2O_2(PSS)/H_2O_2(measurements)$ ratios are shown. The data are binned into subsets

of 1 km of altitude with respect to the center of the bin width based on take-off and landings in proximity to the base of operations in Sal, Cape Verde. The lines and the shadings represent mean values and the 1σ-standard deviations. Dashed lines represent median values.





**Figure 6. Vertical profiles of observed (red), simulated (blue), and calculated based on the PSS assumption (black) hydrogen peroxide means and medians (top panel; a-b) and vertical profiles of mean and median H₂O₂(EMAC)/H₂O₂(measurements) and H₂O₂(PSS)/H₂O₂(measurements) ratios (bottom panel; c–d). Vertical profile estimations were calculated within 1 km means and medians over the atmospheric column based on the data obtained in the region in proximity to the base of operations in Sal, Cape Verde (approx.16°35'–16°51'N; -22°52'–-23°E).**





In general, the observed, PSS-calculated and EMAC-simulated vertical profiles of hydrogen peroxide follow the expected trend throughout the troposphere (Fig. 6a–b). The lowest hydrogen peroxide mixing ratios of 0.141 $ppb_v$ were measured in the upper troposphere, where the availability of the $H_2O_2$ precursor $HO_2$ is limited due to low water vapor concentrations. The highest mean values ($\pm 1\sigma$) of 2.44 ($\pm$ 0.78) $ppb_v$ were measured directly above the boundary layer (2–5 km) in the free troposphere. Below 2 km, the levels of observed $H_2O_2$ decrease to 1.7 ($\pm$ 1.1) $ppb_v$, reflecting the impact of deposition processes

on $H_2O_2$ in the boundary layer in proximity to Cape Verde. Based on the good agreement of the observed vertical distribution with the expected trend as well as log book reports, the presence of clouds and their subsequent scavenging is assumed to have a minor impact on the local budget of the species.

A good agreement between the measured and EMAC-simulated datasets with a $H_2O_2$(EMAC)/$H_2O_2$(measurement) ratio of approximately 1 was found in the lower troposphere (2–6 km; Fig. 6c). Model results tend to overestimate hydrogen peroxide

in the boundary layer, which might be due to the model resolution (1.8° x 1.8°) and the corresponding restrictions in resolving small-scale variations in hydrogen peroxide deposition processes in proximity to the base of operations on the island. The vertical profiles of the observations and the model show that the differences arise mainly in the upper troposphere, with increased ratios of simulated vs. measured $H_2O_2$ of up to a factor 5.6 at 12 km altitude, which might indicate convective outflow in the model. Vertical profiles of observed and simulated $HO_2$, OH, and $H_2O_2$ photolysis rates are in excellent agreement (Fig.

S6), indicating that the model accurately simulates photochemical processes throughout the troposphere, so that the remaining differences for $H_2O_2$ are most likely caused by physical processes (e.g., deposition and transport).

The comparison of measured and PSS-calculated $H_2O_2$ vertical profiles indicates a missing source or an overestimated sink below altitudes of 5 km (Fig. 6b). Here, the PSS-calculated hydrogen peroxide levels fall short by about 1 $ppb_v$ at altitudes below 4 km, yielding $H_2O_2$(PSS)/$H_2O_2$(measured) ratios smaller than unity at these altitudes (Fig. 6d). The absolute difference

between the measured and calculated mixing ratios seems to be very prominent in and right above the boundary layer (< 5 km) and can be associated with air masses affected by Saharan dust, which was often probed during take-off and landings at Sal. Heterogeneous loss of $HO_2$ on desert dust particles, while modest, is expected to lower the production of $H_2O_2$ (Reus et al., 2005).Thus, local mixing ratios of $HO_2$ could be affected by heterogeneous loss, while PSS is not yet achieved (Fig. S7).

In order to investigate the extent of the potential hydrogen peroxide injection into the upper troposphere, we calculate excess

hydrogen peroxide mixing ratios as the difference between the observed and the corresponding $H_2O_2$ based on PSS. Analogously, potential excess of $H_2O_2$ using model-simulated data was determined. The spatial distribution of the calculated transport rates in the upper troposphere is presented in Fig. 7 as latitude vs. longitude plots of mean hydrogen peroxide levels aggregated over a spatial 1° x 1° grid at altitudes above 8 km. The color scale represents the average excess mixing ratios determined for the species in $ppt_v$.




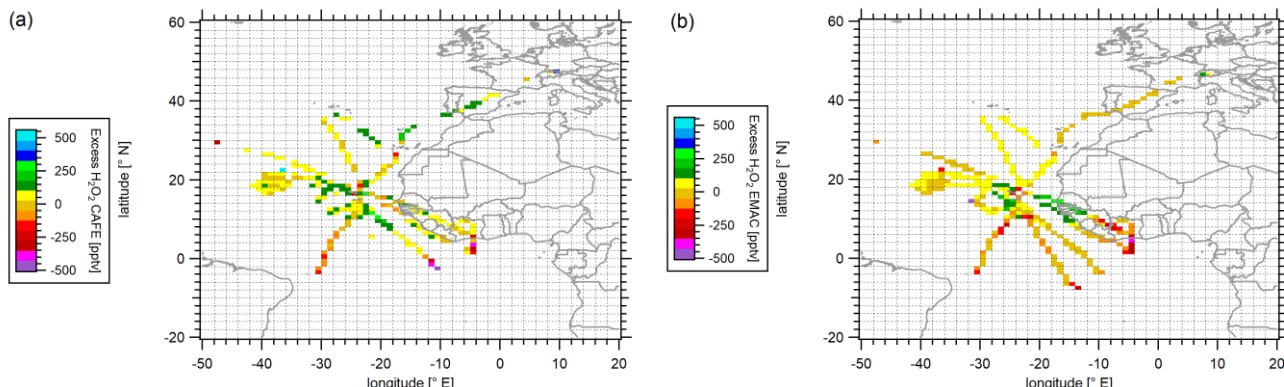


**Figure 7. Color-coded spatial resolution of calculated H$_2$O$_2$ excess based on the difference between the observations and PSS-based calculations (a) and EMAC simulations and calculations at steady state based on the EMAC data output (b). Data were binned into 1° x 1° bins over the full extension of the flight tracks.**

The H$_2$O$_2$ observations exceed the calculated values based on PSS mostly in the range of 70 ppt$_v$ to 110 ppt$_v$ with the exception

of the values derived in the southern hemisphere. Highest deviations were derived in the ITCZ and reach up to 310 ppt$_v$ at their maximum. The excess mixing ratios show a clear trend with the most impactful injection events in the ITCZ region (5°N–15°N), where convective transport is expected (Waliser and Gautier, 1993; Fontaine et al., 2011). From there, a subsequent redistribution of hydrogen peroxide towards the northern and southern hemispheres occurs, which agrees well with the decreasing gradient towards the north and south. EMAC simulates exceeding H$_2$O$_2$ mixing ratios mostly in the range of 20

ppt$_v$ to 150 ppt$_v$, with maximum excess up to 240 ppt$_v$ in the ITCZ (12.5°N, -25.5°E; Fig. 7b). EMAC reproduces the transport rates to a lower extent in the ITCZ, but local convective events such as those occurring within tropical storm Florence and above the African coast are not simulated by the model. In contrast, no significant excess of H$_2$O$_2$ was determined between 1.5°N, - 4.5°E.

The calculated absolute difference between the measured and the PSS-calculated H$_2$O$_2$ in the upper troposphere displays an

average excess of 44 (± 120) ppt$_v$ hydrogen peroxide over the entire region relative to the PSS-modelled conditions. In comparison, injections based on EMAC simulations show about 60% lower values of 18 (± 120) ppt$_v$. The mixing ratios of the H$_2$O$_2$ measurements below 4 km at Sal were in the range of 1.7–2.4 ppb$_v$, which would contribute with up to 1.8–2.6% of H$_2$O$_2$ in the outflow, assuming potential inflow below 4 km. A similar calculation based solely on EMAC data indicates a contribution of 0.6–0.8 % within the model (based on EMAC mixing ratios of 2.1–3.0 ppb$_v$). Thus, although EMAC reproduces

potential transport processes from the lower troposphere, the discrepancy in the transmission efficiencies indicates a smaller contribution to the simulated hydrogen peroxide levels. Consequently, the enhanced hydrogen peroxide mixing ratios cannot be justified exclusively by photochemical reactions within the upper troposphere. Additional injections from the lower troposphere via convective transport and the subsequent redistribution towards the subtropics have to be considered.



## 5    Conclusions

Hydrogen peroxide was measured during the CAFE-Africa campaign over the tropical Atlantic and West Africa in the upper troposphere (above 8 km). Generally, the measured levels of hydrogen peroxide in the upper troposphere fit well to the previously observed $H_2O_2$ at latitudes 10°S–40°N. At high altitudes, a minor symmetrical decrease from the ITCZ towards northern and southern latitudes was observed, which deviates from previously reported observations in the upper troposphere. According to previous reports, the $H_2O_2$ mixing ratios are expected to be elevated in the equatorial upper troposphere due to

biomass burning and atmospheric transport. However, the $H_2O_2$ mixing ratios measured during the CAFE-Africa campaign show only very little latitudinal variation relative to previous measurements over the Atlantic with a shift of the maximum mixing ratios towards the ITCZ. The measured hydrogen peroxide mixing ratios show a rather uniform distribution with peak events in the ITCZ and over the African coast, indicating the influence of convective transport processes on the distribution of hydrogen peroxide in the upper troposphere.

Whilst the observations of hydrogen peroxide are in good general agreement with the range of previous observations performed in the upper troposphere, the measured $H_2O_2$ mixing ratios deviate from the PSS calculations based on OH and $HO_2$ measurements and the simulations performed by EMAC. The local PSS calculations significantly underestimate the $H_2O_2$ mixing ratios in the north of the probed region. There, the comparison of the $H_2O_2$ measurements with PSS calculations reveals a large impact of vertical transport within the ITCZ and the associated redistribution in the upper troposphere on the spatial

distribution of hydrogen peroxide. Further, the enhanced presence of clouds in the ITCZ and towards the southern hemisphere indicates significant cloud scavenging in the probed region, justifying the deviations to the local photostationary steady-state calculations, which only account for photochemical sources and sinks of $H_2O_2$. The EMAC simulations of $H_2O_2$, $HO_2$, and OH agree with the observations in the lower tropospheric layers. An overestimation of model results compared with the observations of hydrogen peroxide mixing ratios due to inaccuracies in cloud scavenging was observed in the upper

troposphere towards the southern hemisphere. Based on our calculations, the model simulates only partially the impacts of atmospheric transport on the $H_2O_2$ budget. In fact, the calculated excess hydrogen peroxide mixing ratios based on EMAC are lower compared to those based on the PSS calculations by approximately 60%. The comparison between the EMAC- and PSS-calculated versus measured hydrogen peroxide confirms that convective transport and consequent redistribution most likely by northerly and southerly winds towards the subtropics has a significant impact on $H_2O_2$. This redistribution alters the spatial

distribution of $H_2O_2$ towards more uniform mixing ratios in the marine tropical upper troposphere than would be expected based exclusively on photochemical production and loss processes in the UT.

**Data availability.** All CAFE-Africa data sets used in this study are permanently stored in an archive on the KEEPER service of the Max Planck Digital Library (https://keeper.mpdl.mpg.de; last access: 17 October 2022) and are available to all scientists,

who agree to the CAFE-Africa data protocol.



**Author contributions.** JL and HF planned the campaign; DD, BBr, RR, DM, MM, HH, and BB performed the measurements; ZH and HF designed the study; ZH, DD, RR, BB, and MZ processed and analyzed the data; AP developed the model code and performed the simulation; ZH wrote the manuscript draft with contributions of all co-authors.


**Competing interests.** The authors declare that they have no conflict of interest.

**Acknowledgments.** The authors are very grateful to the CAFE-Africa team, Forschungszentrum Jülich, Karlsruhe Institute of Technology and Deutsches Zentrum für Luft- und Raumfahrt (DLR) in Oberpfaffenhofen for their great support. Their work
was essential for the project.

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
