# Peer review of "Hydrogen peroxide in the upper tropical troposphere over the Atlantic Ocean and western Africa during the CAFE-Africa aircraft campaign"

_Atmospheric Chemistry and Physics, 2022_

## Author Comment (AC1)

**Please note the used color code**
**(black: Referee Comments, red: Author Comments, blue: manuscript changes according to Referee's recommendations and comments)**

This paper presents $H_2O_2$ observations in the upper troposphere over the tropical Atlantic from the CAFE campaign and compares to photochemical steady state (PSS) values and to the EMAC model.

Overall assessment: The observations are interesting but the interpretation is routine, and as a result I don't see any significant scientific advances coming out of this paper. This seems a missed opportunity because the observations could be used to lend new insights into several interesting questions: (1) the scavenging efficiency of $H_2O_2$ in deep convection, (2) whether $H_2O_2$ is directly injected or is produced following the injection of $CH_3OOH$, and (3) what time scales are involved in the evolution from fresh convective injection to PSS. These are questions that have been debated in the literature and for which this data set (it seems to me) could provide new answers. Instead, the comparison to model results is long and tedious but does not go beyond being descriptive and anecdotal, and I finished this paper without the impression of having learned anything. The PSS calculation is wrong – $H_2O_2$ has too long a lifetime to be in instantaneous PSS (it needs to be 24-h PSS). The convective influence on $H_2O_2$ in the upper troposphere is by itself not new – this has been shown in several previous aircraft campaigns.

I'm sorry to be so negative because I feel that the observations are valuable and could provide the basis for a good paper.

We thank the referee for the assessment and the helpful comments. The work gives a general overview of $H_2O_2$ levels over the tropical Atlantic and the impact of convective processes taking place in ITCZ on the total budget and the spatial distribution of hydrogen peroxide in the upper troposphere. To our knowledge, tropical hydrogen peroxide measurements and airborne-based analyses at altitudes above 12 km are scarce due to technical restrictions associated with aircraft measurements.

In our work, we show that additional $H_2O_2$ is not only injected into the UT (above 8 km) in the convection prone region of the ITCZ but further subsequently redistributed towards the subtropics. For this reason, other than in the free troposphere and upper troposphere below 12 km, which was shown during previous airborne studies, the levels of hydrogen peroxide do not display any significant latitudinal gradient corresponding with the inclination of the solar radiation. In our studies, we show that the levels of $H_2O_2$ seem to be far more dependent on additional sources than previously assumed.

We thank the referee for the valuable suggestions on analyses, which might provide significant insight to highly interesting studies on deep convection processes. As described in Sect. 2, the objective of the mission was to investigate large-scale distribution of trace gases, radicals and aerosols in the upper tropical troposphere. As displayed in the three-dimensional $H_2O_2$ distribution below, unfortunately only few vertical profiles were measured outside of the region of the base of operation on Cape Verde and the ITCZ. We believe that, due to a lack of statistical significance in other vertically performed flight legs, only the 30 take-offs and landings give sufficient information on the distribution of hydrogen peroxide throughout the tropospheric column. For this reason, based on the available dataset, the scavenging efficiency in deep convection and insights on injection processes and time scales cannot be considered, unfortunately.

As further described in the experimental Sect. 3.1, due to the instrumental technique, detailed characterizations of single organic peroxide species are not possible. The measurements obtain the total amount of organic hydroperoxides without distinction between the organic species in the sampled air. Thus, the measurement uncertainty of hydroperoxides other than $H_2O_2$ is about 40%. We believe, this

high uncertainty factor does not serve well for any quantitative analyses on the organic hydroperoxide budget in the upper troposphere as well as on any related questions regarding convective transport.

The presented calculations of hydrogen peroxide under photostationary steady-state conditions are, as mentioned throughout the work, of local character and serve exclusively the characterization of atmospheric regions, where observed hydrogen peroxide levels deviate from the expectations due to potential additional sources and sinks (e.g. injections due to deep convection, cloud scavenging and rainout). We do not intend to compare the observations of hydrogen peroxide with PSS calculations beyond the aspect of the observed local imbalances between photochemical production and loss paths of the species.

We hope the following clarifications and comments will improve the overall understanding of the analyses and results, which we are presenting in the scope of this work.

[Figure]

Figure 1: General overview of performed measurement flights with respect to the observed $H_2O_2$ levels (color-code) with the assumed region of ITCZ during the campaign highlighted in grey.

**Specific comments (lines)**

40-45, also 68-69: this summary of $H_2O_2$ chemistry is textbook wrong. $HO_2$ does not necessarily come mainly from CO+OH; any VOC+OH (for example $CH_4$+OH) will lead to $HO_2$. And the 'competition' between self-reaction of $HO_2$ and reaction with NO and $O_3$ (producing OH) does not actually affect $H_2O_2$ formation in a $NO_x$-limited regime because the OH will go back to $HO_2$.

We agree with the referee that in general, the most organic peroxy radicals also form $HO_2$ in secondary reactions Also, other primarily production paths, as in case of formaldehyde photolysis, might contribute to $HO_2$ production. Nevertheless, under the UT conditions, which were investigated in this work, $HO_2$ is expected to be mainly formed via reaction of carbon monoxide with OH radicals. Further, as shown by Tadic et al. (2021), the production rate of $HO_2$ via reaction of OH is on average a factor 5 higher than the production rate via photolysis of HCHO.

It is correct that the OH formed via the reaction of $HO_2$, NO and $O_3$ might regenerate $HO_2$. Nonetheless, in an $NO_x$ rich environment both, OH and $HO_2$, also react with NO and $NO_2$ to form $HNO_3$ and $HNO_4$, the latter being mostly abundant in the upper troposphere. Thus, the literature (e.g. Reeves and Penkett 2003) reports about competition between these reactions.

Lines 40 – 45 (now L44 – 49) and L68 – 69 (now L74 – 76) were changed according to the referee's comment.

L44 – 49 (former L40 – 44): The most prominent pathway leading to $H_2O_2$ is the self-reaction of $HO_2$ radicals (R4). $HO_2$ can be formed by many pathways. Under the upper troposphere conditions investigated in this work, $HO_2$ is mainly formed via reaction of carbon monoxide with OH radicals which can be formed initially in the photolysis of ozone and the subsequent reaction of $O^1D$ with water vapor (R1-R3). Moreover, OH can be recycled from $HO_2$ in reactions with NO or $O_3$. The budget of $H_2O_2$ is thus controlled by the steady-state concentration of $HO_2$ radicals and the main gas-phase loss processes photolysis and reaction with OH (R5-R6).

L74 – 76 (former L68 -69): In urban areas, the formation of hydrogen peroxide is diminished by the increased mixing ratios of $NO_x$ ($NO_x = NO_2 + NO$) derived from anthropogenic sources, as the self-reaction of $HO_2$ to $H_2O_2$ is competing with the much faster reaction of $HO_2$ with NO (Lee et al., 2000; Reeves and Penkett, 2003).

126: the instrument measures ROOH as well as $H_2O_2$, and I would have expected the ROOH data to be brought into the analysis, in particular as diagnostic of fresh convective injection and chemical aging in the upper troposphere. This seems like a missed opportunity.

The ROOH data are measured as a sum of the different species, and a detailed characterization of each sampled organic peroxide compound using the measurement technique is not possible. Given a TMU of 40% as stated in the work, the data lacks further relevance to reliably quantify any convective injection or chemical aging of hydrogen peroxide in the upper troposphere.

201: $H_2O_2$ has a lifetime of days and therefore cannot be assumed to be in instantaneous PSS. PSS would have to be calculated over a 24-h diurnal cycle with periodic boundary conditions.

We agree with the referee on the extended $H_2O_2$ lifetime in the troposphere. Nonetheless, as discussed above, the presented calculations under PSS conditions are of local character and serve exclusively to identify the atmospheric regions, where observed hydrogen peroxide levels deviate from the expectations.

207: what would be the 'other gases producing $H_2O_2$'? According to current knowledge $HO_2+HO_2$ is the only source.

We apologize for the confusion. Former L207 was removed from the work according to the referee's comment.

222: what does 'supplement' mean? It is not clear to me if $HO_2$, OH, and $JH_2O_2$ come from the measurements or from the model.

We apologize for the confusion due to the wrong choice of words. The results generated by the model EMAC were compared with the observations in order to analyze the accuracy of the model's performance, especially regarding non-photochemical processes in the troposphere.

L236 (former L221): For the purpose of the present study, we used measured $H_2O_2$, OH, $HO_2$, water vapor, $j(H_2O_2)$, temperature, and pressure and compare these with the concurrent spatially interpolated EMAC simulations. To synchronize the time resolution of the simulated data with the measurement output, we calculated a mean of the measurement data with a matching temporal resolution of 6 min (equivalent to model output).

244: why continental? Couldn't it be marine?

L259 (former L244) was changed according to the referee's comment.

L259 (former L244): These maxima are most likely due to atmospheric transport of $H_2O_2$ into the upper troposphere.

259-263: SONEX was also in the fall when $H_2O_2$ would be lower. The authors may be right that the higher values in ATom (not just 'slightly') could be due to lower-altitude sampling but that would affect the other campaigns as well that used the NASA DC-8. It would be good to show the vertical profiles (Figure 6) earlier in the paper to make that point.

We thank the referee for the comment. Since, as mentioned in Sect. 3.5, the vertical profiles are restricted to the take-offs and landings at the base of operation at Sal, the shown vertical trends serve rather the purpose of the comparison between the observations and the EMAC simulations as well as to point out in detail, where deviations from the PSS are expected in the sampled region at the base of operation. Surely the trends in deviations are expected to be quite different especially in the southern part of the domain, where the air masses are known to be highly impacted by biomass burning emissions, as can be seen in the plots accounting the whole data set (e.g. below 8 km of altitude).

267-268: 'hydrogen peroxide in the upper troposphere seem to be far less dependent on latitude than those at lower altitudes'. I don't see the evidence for this. The lack of latitudinal gradient here is likely because the campaign was in summer.

As described in Sect. 2, the measurement flights were performed between August and September, not in the summer alone. Additionally, the campaign took place over the tropical Atlantic, which by means of the region and the meteorology does not show important seasonal dependencies. The main seasonally driven difference would apply to measurements in the boundary layer and the free troposphere in the southern hemisphere due to differences in biomass burning emissions. Nonetheless, as the works clearly shows, there is no evidence of biomass burning emissions affecting hydrogen peroxide levels in the upper troposphere in the ITCZ and towards the subtropics.

291: shouldn't 'decreasing' be 'increasing'?

We apologize for the confusion. L307 (former L291) changed according to the referee's comment.

L307 (former L291): With decreasing latitude, the model tends to significantly overestimate hydrogen peroxide; $H_2O_2$(EMAC)/$H_2O_2$(measurement) ratios are increasing from approximately 2 to 4 with decreasing latitude.

293: why would EMAC underestimate cloud scavenging? That's generally not considered an issue in the upper troposphere where precipitation is infrequent.

Fig. 4 as well as in Fig. S2, present the respective information regarding the presence of clouds in the upper troposphere. Since generally the clouds are highly variable and complex and may consist of ice and supercooled liquid water, an uptake of hydrogen peroxide in all type of clouds should be assumed. In fact, to our knowledge, there is so far no evidence for hydrogen peroxide scavenging exclusively within the rain or liquid water phases of the clouds. $H_2O_2$ could be temporarily or permanently absorbed by ice particles in clouds as well.
As discussed further in the work L379 – 390 (former 365 – 384) and L426 – 428 (former L414 – 416) due to the rather minor differences between the measured and simulated $H_2O_2$ precursors (HO$_2$, OH and j($H_2O_2$)), we assume the chemistry and photolysis not to be the major reason for the deviations between the model and the observations. From our past analyses we have learned that EMAC has difficulties to accurately simulate cloud scavenging (Klippel et al. 2011; Hottmann et al. 2020; Hamryszczak et al. 2022).

L309 (former L293) was changed.

L309 (former L293): Locally, most likely due to underestimated cloud scavenging as will be further discussed in this work, EMAC simulates highly elevated hydrogen peroxide with a factor of up to 14 in excess of the measurements (4.5°N, -9.5°E).

389: the departure from PSS seems to be only above 12 km. Is this because deep convective outflow was above that altitude? It seems from the comparison with the EMAC vertical profile that EMAC may release outflow at 10-12 km and thus underestimate the depth of tropical convection, which is a common problem in models.

That is correct. As discussed in L423 – 426 (former L411 – 414), the increased $H_2O_2$(EMAC)-to-$H_2O_2$(measurement) ratio, which is based on observed and simulated $H_2O_2$ levels, might indicate the simulated depth of the convective outflow in the model at 10 – 12 km.

404-405: if deposition is important below 2 km, how come PSS underestimates observations there?

As discussed in the work (L429 – 435, former L417 – 423), the comparison of the vertical profiles indicates a missing source or an overestimated sink below altitudes of 5 km. The difference between the measured and calculated mixing ratios can be associated with air masses affected by Saharan dust, which were often sampled during take-offs and landings at Sal. Due to deposition of $HO_2$ on desert dust particles, the production term of PSS-$H_2O_2$ (Eq. 2) is affected and thus, the PSS calculation underestimates the levels of hydrogen peroxide relative to the observations (Eq. 6).

423: I thought $HO_2$ was directly measured? In any case, this dust uptake explanation would not help explain the PSS underestimate in the MBL.

As described in the experimental Sect. 3.2, $HO_2$ was measured using a laser-induced fluorescence-based method (LIF; additional chem. conversion for $HO_2$) with the instrument HORUS. Fig. S7 presents a direct comparison between the measured $HO_2$ and PSS-based $HO_2$ calculated using the observed levels of $H_2O_2$. As stated above, $HO_2$ was most likely removed by deposition on dust particles, which affects the production term of PSS-$H_2O_2$ (Eq. 2, 6). Based on the decreased mixing ratios of $HO_2$ up to 5 km altitude relative to the PSS-$HO_2$ a direct correlation with the underestimation of $H_2O_2$ under PSS conditions can be assumed. Here, the production of $H_2O_2$ is underestimated relative to the measurements due to decreased levels of $HO_2$.

431: Figure 7 doesn't seem to add anything.

Figure 7 displays the local absolute amount of excess or lost $H_2O_2$. The presented spatial resolution indicates the imbalance between the sources and sinks and the respective regions, where processes of non-photochemical origin are impacting the budget of $H_2O_2$. The calculated excess hydrogen peroxide levels quantify the impact of transport processes due to convection in the ITCZ and due to the subsequent redistribution towards the subtropics. Further the estimated levels show the amount of $H_2O_2$, which is missing due to cloud scavenging on a local scale, where enhanced presence of clouds was observed (Fig. S2).

**References**

Hamryszczak, Zaneta Teresa; Pozzer, Andrea; Obersteiner, Florian; Bohn, Birger; Steil, Benedikt; Lelieveld, Jos; Fischer, Horst (2022): Distribution of hydrogen peroxide over Europe during the BLUESKY aircraft campaign.

Hottmann, Bettina; Hafermann, Sascha; Tomsche, Laura; Marno, Daniel; Martinez, Monica; Harder, Hartwig et al. (2020): Impact of the South Asian monsoon outflow on atmospheric

hydroperoxides in the upper troposphere. In *Atmos. Chem. Phys.* 20 (21), pp. 12655–12673. DOI: 10.5194/acp-20-12655-2020.

Klippel, T.; Fischer, H.; Bozem, H.; Lawrence, M. G.; Butler, T.; Jöckel, P. et al. (2011): Distribution of hydrogen peroxide and formaldehyde over Central Europe during the HOOVER project. In *Atmos. Chem. Phys.* 11 (9), pp. 4391–4410. DOI: 10.5194/acp-11-4391-2011.

Reeves, Claire E.; Penkett, Stuart A. (2003): Measurements of peroxides and what they tell us. In *Chemical reviews* 103 (12), pp. 5199–5218. DOI: 10.1021/cr0205053.

Tadic, Ivan; Nussbaumer, Clara M.; Bohn, Birger; Harder, Hartwig; Marno, Daniel; Martinez, Monica et al. (2021): Central role of nitric oxide in ozone production in the upper tropical troposphere over the Atlantic Ocean and western Africa. In *Atmos. Chem. Phys.* 21 (10), pp. 8195–8211. DOI: 10.5194/acp-21-8195-2021.

---

## Author Comment (AC2)

**Please note the used color code**
**(black: Referee Comments, red: Author Comments, blue: manuscript changes according to Referee's recommendations/comments)**

This paper presents results from the CAFÉ-Africa aircraft mission, focusing on the measurements of hydrogen peroxide (HOOH). HOOH is an important species that controls or is an indicator of the oxidative capacity of the troposphere. The authors compare the observed HOOH with global chemistry models and with photochemical calculations using the observations made during the flights. This is an important paper for the observations, but the model-measurement evaluation is confusing and not insightful. There are a lot of confusing comparisons and speculation. It needs some cleanup. The authors should come up with one or two major points and support them.

We thank the referee for her/his assessment. Following the referee's recommendation, the manuscript was changed as described below.

What is missing here is some idea of the variability of HOOH in the upper troposphere. We are shown 6 min averages binned into 2-degree latitude blocks (Fig. 3). The figure below shows all the ATom measurements of HOOH (from the 10s merged data) as a function of latitude. These are ~10,000 points and only include data above 8 km (comparable to the CAFÉ data here). The mean/median value of the points is 105/82 ppt. This is consistent with CAFÉ results (Fig. 2) but the large scatter, but with many (>10%) above 200 ppt indicates recent convective sources at almost all latitudes at some time or another (warning, this is all the 4 ATom deployments). It is hard to see a latitudinal pattern except for the much higher baseline values between 10S and 20N).

**Figure:** HOOH (ppt) vs. latitude. ATom-1234, above 8 km, Atlantic basin

We thank the reviewer for her/his helpful comments. We apologize for not noticing promptly that the figure, which the reviewer is referring to (HOOH (ppt) vs. latitude. ATom-1234, above 8 km, Atlantic basin), is not displayed. Thus, we cannot follow the referee's comments on the ATom results regarding the species spatial distribution in the upper troposphere, unfortunately.
As stated in Sect. 3.5., for all spatially resolved data based on the observations, PSS-model calculations and EMAC simulations were calculated as **1° x 1°** bins over the full extension of the sampled regions in the upper troposphere (e.g. 8 km or higher; Fig. 2 & 3). Given the average aircraft speed of 232 m/s in the upper troposphere and the instruments time resolution of 2 min, the average of 3.6 points per 100 km and thus an approx. average of 12 – 13 points per 1° x 1° bin can be assumed. Further, the observation-based data was binned into 6 min averages for the purposes of the study and the comparison with the model EMAC, of which the temporal resolution is restricted to 6 mins, as described in Sect. 3.3. and 3.5. Following the calculation presented above, this gives approximately 1.4 points per calculated bin of 1° x 1°. Thus, the variability the referee is missing in the $H_2O_2$ distribution is due to the observed levels, or calculations derived from these, and not due to the process used to calculate and display the spatial resolution of $H_2O_2$ during the campaign.

Details:

L15: I would not describe this as a "uniform latitudinal distribution" from either Fig 3 or the ATom data. There are clear hot spots at several latitudes. Admittedly there is no clear latitudinal gradient from 40S-40N.

L15 was changed according to the recommendation.

L15: The measured hydrogen peroxide mixing ratios in the upper troposphere show no clear trend in the latitudinal distribution with locally increased levels (up to 1 ppbv) within the Intertropical Convergence Zone (ITCZ), over the African coastal area, as well as during measurements performed in proximity of the tropical storm Florence (later developing into a hurricane). The observed $H_2O_2$ distribution suggests that mixing ratios in the upper troposphere seem to be far less dependent on latitude than assumed previously and the corresponding factors influencing the photochemical production and loss of $H_2O_2$. The observed levels of $H_2O_2$ in the upper troposphere indicate the influence of convective transport processes on the distribution of the species not only in the tropical but also in the subtropical regions.

L19-20: This is too simple and wrong. I see in Fig 3 values > 1 at 25N and 30N. Likewise values <<1 are scattered all over including near the ITCZ (which is not that well defined). The question is: Are all the ratios > 1 associated with recent deep convection. Your pattern in Fig 3 seems patchy.

We agree with the referee on the presence of ratio values > 1 north from the ITCZ and << 1 in the ITCZ, Ratios above 1 indicate that relative to PSS calculations, the observations tend to display lower $H_2O_2$ mixing ratios. This implicates an additional sink in the hydrogen peroxide budget relative to the photostationary steady-state, which is most probably not related to photochemical processes in the troposphere.
L19 – 20 (now L22) were changed according to the referee's comments on the presence of enhanced $H_2O_2$(PSS)-to-$H_2O_2$(Obs) in the North of the ITCZ.

L22 (former L19 -20): North of the ITCZ, PSS calculations produce mostly lower $H_2O_2$ mixing ratios relative to the observations. The observed mixing ratios tend to exceed the PSS calculations by up to a factor of 2. With the exception of local events, the comparison between the calculated PSS values and the observations indicates enhanced $H_2O_2$ mixing ratios relative to the expectations based on PSS calculations in the North of the ITCZ.

L21: are there not some <1 squares here?

L21 was changed according to the referee's remark.

L26 (former L21): On the other hand, PSS calculations tend to overestimate the $H_2O_2$ mixing ratios in most of the sampled area in the south of the ITCZ by a factor of up to 3.

L23: You talk about scavenging of HOOH but there is no evidence here to show it. What I read is that deep convection brings up high values of HOOH that are above the local PSS values, and that over time (How much time??) these go back to PSS. What this work does not explain is how this ratio drops below 1 for many palaces. Scavenging is not possible as there is no liquid water clouds up here. Perhaps our PSS model is wrong?

As shown in Fig. 4 as well as in Fig. S2 (top panel) based on ERA5 reanalyses, the presence of clouds in the upper troposphere cannot be neglected. Since generally the clouds are highly variable and complex and can involve ice and supercooled liquid water, we can assume the uptake and the scavenging of hydrogen peroxide in the sampled area affected by the presence of clouds. To our understanding there is so far no evidence for hydrogen peroxide scavenging exclusively by rain or liquid water phases of the clouds, $H_2O_2$ could be absorbed by ice particles in clouds as well.

L25: I am not sure the "north and south" here is correct, the spreading of convective outflow could be east-west also.

Former L25 (now L29) was removed from the work.

L29: I would not call this 'uniform' but rather that it showa no clear gradient with latitude.

Former L29 (now L32) was changed according to the referee's comment.

L32: In contrast, the measurements show no clear gradient with latitude in the mixing ratios of $H_2O_2$ in the upper troposphere with a slight decrease from the ITCZ towards the subtropics, indicating a relatively low dependency on the solar radiation intensity and the corresponding photolytic activity.

L30: I do not see how you can attribute this just to underestimated loss instead of overestimated production. The lifetime of HOOH up here is how long (about 2 days?) and you calculate PSS values based on instant measurements – what is the error/uncertainty in this? Your comparison with PSS misses out on the transient response.

We use the local PSS calculations exclusively to identify the regions that present either additional sources or sinks of hydrogen peroxide or are not homogeneous with respect to transport. As presented in Sect. 4.2 and specifically in Fig. 4 and L330 – 337 (former L314 – 321), the highest deviations correspond with underestimated $H_2O_2$ loss due to enhanced cloud presence, the subsequent scavenging in the ITCZ and towards the South.

Abstract: Overall, the abstract could be improved, and made to summarize the facts of the observations and then the surmises of the authors when comparing with models (PSS or EMAC).
I cannot easily understand what the authors have discovered from CAFE-Africa from this abstract.

The abstract was changed according to the referee's comments.

**Abstract.** This study focuses on the distribution of hydrogen peroxide ($H_2O_2$) in the upper tropical troposphere at altitudes between 8 and 15 km based on *in situ* observations during the Chemistry of the Atmosphere – Field Experiment in Africa (CAFE-Africa) campaign conducted in August–September 2018 over the tropical Atlantic Ocean and western Africa. The measured hydrogen peroxide mixing ratios in the upper troposphere show no clear trend in the latitudinal distribution with locally increased levels (up to 1 ppbv) within the Intertropical Convergence Zone (ITCZ), over the African coastal area, as well as during measurements performed in proximity of the tropical storm Florence (later developing into a hurricane). The observed $H_2O_2$ distribution suggests that mixing ratios in the upper troposphere seem to be far less dependent on latitude than assumed previously and the corresponding factors influencing the photochemical production and loss of $H_2O_2$. The observed levels of $H_2O_2$ in the upper troposphere indicate the influence of convective transport processes on the distribution of the species not only in the tropical but also in the subtropical regions. The measurements are compared to observation-based photostationary steady-state (PSS) calculations and numerical simulations by the global EMAC model. North of the ITCZ, PSS calculations produce mostly lower $H_2O_2$ mixing ratios relative to the observations. The observed mixing ratios tend to exceed the PSS calculations by up to a factor of 2. With the exception of local events, the comparison between the calculated PSS values and the observations indicates enhanced $H_2O_2$ mixing ratios relative to the expectations based on PSS calculations in the North of the ITCZ. On the other hand, PSS calculations tend to overestimate the $H_2O_2$ mixing ratios in most of the sampled area south of the ITCZ by a factor of up to 3. The significant influence of convection in the ITCZ and the enhanced presence of clouds towards the southern hemisphere indicate contributions of atmospheric transport and cloud scavenging in the sampled region. Simulations performed by the EMAC model also overestimate hydrogen peroxide levels particularly in the southern hemisphere, most likely due to underestimated cloud scavenging. Both, EMAC simulations and PSS calculations indicate a latitudinal gradient from the equator towards the subtropics. In contrast, the measurements show no clear gradient with latitude in the mixing ratios of $H_2O_2$ in the upper troposphere with a slight decrease from the ITCZ towards the subtropics, indicating a relatively low dependency on the solar radiation intensity and the corresponding photolytic activity. The largest model

deviations relative to the observations correspond with the underestimated hydrogen peroxide loss due to enhanced cloud presence, scavenging, and rainout in the ITCZ and towards the south.

L50: where is the heterogeneous cloud loss (wet scavenging at least?)

The presented photochemical reactions summarize the main processes leading to the production and loss of hydrogen peroxide in the troposphere. A short elaboration on physical losses due to clouds and deposition can be found in L59 – 61 (now L64 – 66).

L56: the missing question is what is the $HO_x$ source > 8 km, OH is not mainly from HOOH, or is it? We know it is not acetone any more, but what else? $O^1D + H_2O$ is operating, but ore slowly.

We thank the referee for pointing out this very important question. The production of OH might be dependent to an extent on the HCHO production due to $HO_x$ recycling via more efficient $HO_2$ to OH conversion by NO, which can be produced via lightning assimilated with deep convection in the upper troposphere. L78 (now L85) was changed according to the referee's comment.

L78 (now L85): Thus, convective processes are expected to contribute to increased levels of hydrogen peroxide in the upper troposphere and promote elevated $HO_x$ levels via subsequent photochemical processes involving $H_2O_2$ degradation as well as HCHO production due to efficient $HO_x$ recycling via the reaction with NO produced by lightning during the convective episodes (Jaeglé et al., 1997; Jaeglé et al., 2000; Nussbaumer et al. 2021; Tadic et al. 2021).

L59: not all wet scavenging leads to deposition, it can be released by virga, e.g.

We agree with the referee. Cloud scavenging can be of temporal (transport, release) as well as permanent effect (rainout or liquid phase reactions) character. Nonetheless, locally scavenging always leads to a decrease of hydrogen peroxide mixing ratios.

L63a: Agreed, this is what ATom shows for the tropical Atlantic – but in models, part of this comes from the maximum in OH production (and hence $HO_2$) above the MBL (950 hPa) because of clouds (Spivakovsky et al., 2000).

We thank the referee for the comment. Former L61 (now L66) was changed according to referee's comment.

L66 (former L61): In effect, based on the availability of the $H_2O_2$ precursors (OH and hence $HO_2$) and the corresponding photochemical reactions producing and removing $H_2O_2$ in the troposphere as well as on the discussed physical processes, the vertical distribution of $H_2O_2$ often follows an inverted c-shape with decreased mixing ratios within the boundary layer and the upper troposphere and a local maximum in the middle troposphere at altitudes between 2 and 5 km.

L63b: This only applies here to > 8km, and the previous sentence talks about 2-5 km and the MBL.

L70 (former L63b) was adjusted to the referee's comment.

L70: Additionally, observations in the UT (> 8 km) indicate a decreasing trend approximately from the equator towards the north and south (Daum et al., 1990; Heikes, 1992; O'Sullivan et al., 1996; Weinstein-Lloyd et al., 1998; Snow, 2003; Snow et al., 2007; Klippel et al., 2011).

L79: 'might' to 'is expected to…
'

L85 (former L79) was changed according to the recommendation.

L85: Thus, convective processes are expected to contribute to increased levels of hydrogen peroxide in the upper troposphere and promote elevated $HO_x$ levels via subsequent photochemical processes involving $H_2O_2$ degradation as well as $HO_x$ recycling via the reaction with NO produced by lightning during the convective episodes (Jaeglé et al., 1997; Jaeglé et al., 2000; Nussbaumer et al. 2021; Tadic et al. 2021).

L83: correct, what we are missing here is the 'vertical distribution'

We thank the referee. This is correct.

L93: What does 'mean values' here refer to? The comparison with ATom here is odd. Why did you only compare with the Aug and Oct deployments? This is a tropical measurement and the seasonal differences are small and mostly related to variations around ITCZ or major biomass burning layers. I did a quick sampling of the ATom 1234 deployments over the tropical Atlantic above 8 km. There are about 1,000 10s points in each sample. I do not understand how your 'mean values' range from 0.05 to 0.25 ppb. The Allen et all. 2022 paper has mean and median tables, but their values for the Atlantic are much higher because they are not limited to >8km.

| 20S-20N | Atom-1 | ATom-2 | ATom-3 | ATom-4 | Here (all Latitudes |
|---|---|---|---|---|---|
| mean (ppb) | 0.14 | 0.14 | 0.09 | 0.12 | 0.18 |
| median (ppb | 0.12 | 0.12 | 0.07 | 0.11 | 0.15 |

We apologize for the confusion. As it was stated in the discussion (former L94 – 95), due to restricted accessibility of the ATom mean values exclusively above 8 km, the approximate mean values for the UT where obtained from the plots presented in the study. We thank the referee for his/her extra effort on sampling the ATom data over the tropical Atlantic above 8 km of altitude and we would like to extend the comparison in L93 and further discussions by the remaining ATom measurements and the mean and median values presented by the referee with a citation note on the communication (private or namely). L93 (now L100) was changed according to the referee's comment.

L100 (former L93): During the Atmospheric Tomography Misson (ATom) performed in August 2016 (ATom-1), February 2017 (ATom-2), October 2017 (ATom-3) and May 2018 (ATom-4), mean values ranging between 0.09 ppbv up to 0.14 ppbv were measured over the Mid-Atlantic Ocean (20°S–20°N; Allen et al., 2022; [private/namely communication]). Please note that the average values within the upper troposphere cited here are based on exclusively sampling the ATom data over the tropical Atlantic above 8 km of altitude and do not necessarily match the general results over the entire sampled tropospheric column, as presented in the cited work.

L113: 'probed area' is odd English usage. Can you give any info on vertical sampling?

Please find below the vertical profiles of observed, calculated according to the PSS assumption and simulated $H_2O_2$ generated based on the entire dataset.
L124 (former L113) was changed regarding to the referee's recommendation.

L124: The investigated area covered a latitudinal and longitudinal range from approximately 10°S–50°N and 50°W–15°E. The majority of vertical sampling was performed in close proximity of the base of operation and covered the altitudinal range between a few tens of meters above the surface and the maximal flown altitude (15 km). In sum, 30 takeoffs and landings with ascending and descending rates of 900 – 1100 m/min and 450 – 650 m/min, respectively were performed giving an average descend/ascent rate of 775 m/min (with 1 point per 1550 m in vertical sampling at instruments temporal resolution of 2 min).

[Figure]

Figure 1: Vertical profiles of observed (red), simulated (blue), and calculated based on the PSS assumption (black) hydrogen peroxide means and medians. Vertical profiles were calculated for 1 km means and medians over the atmospheric column based on the data obtained in the entire region sampled.

L219: Did you check using your EMAC model if eqn6 really does produce the correct PSS when all reactions and missing species are accounted for. This should be an obvious check.

The mentioned test was performed prior to the analyses presented in this work. Please find attached below the comparison of the local PSS $H_2O_2$ vertical profiles based on the observations and on the simulation output for the entirety of the campaign. There are no statistically significant deviations between the results of both calculations apart from the resolution restricted simulations in the BL, where EMAC has trouble to simulate small scale variations in hydrogen peroxide deposition processes in proximity to the base of operation on Cape Verde, which was discussed in the work in L421 – 423 (former L409 – 411).

[Figure]

Figure 2: Vertical profiles of observation based (black) and EMAC simulation based (light blue) H$_2$O$_2$ under assumption of PSS conditions. Vertical profile estimations were calculated within 1 km means and medians over the atmospheric column based on the data obtained in the entire sampled region.

L221: This is odd. Why use EMAC to supplement observations. Then the comparison is corrupted. You need some independence. Also, Why is the data every 6 min when the HOOH is measured every 122 s = 2 min as stated above. These do not add up.

We apologize for the confusion due to the wrong choice of words. The results generated by the EMAC model were compared with the observations in order to analyze the model performance, especially regarding non-photochemical processes in the troposphere. Due to the model restrictions in the time resolution (only 6 min) and in order to synchronize the time step of the simulated data with the measurement output, we obtained the mean of the measurement data with a corresponding temporal resolution.

L236 (former L221): For the purpose of the present study, we used measured H$_2$O$_2$, OH, HO$_2$, water vapor, j(H$_2$O$_2$), temperature, and pressure and compared these with the concurrent spatially interpolated EMAC simulations. To synchronize the time resolution of the simulated data with the measurement output, we calculated a mean of the measurement data with a matching temporal resolution of 6 min (equivalent to model output).

L241: Are the mean and median stats from the 1x1 degree averages? or from each measurement? This is not clear.

The mean and median calculations are based on the entirety of the performed observations under the consideration of the chosen temporal resolution. These values do not represent spatial averages.
L 256 (former L241) was adjusted to the referee's comment.

L256: The mean (±1σ) and median mixing ratios based on all measured H$_2$O$_2$ mixing ratios during the campaign were 0.18 (± 0.13) ppbv and 0.15 ppbv, respectively, with maximum hydrogen peroxide mixing ratios reaching 1.03 ppbv.

L243 – locally up to 0.67 ppb means what?  You found one 122s observation with 0.67 ppb? [restricted to > 8km).

We apologize for the confusion. The locally mentioned high levels are based on the data binned into 1° x 1° bins over the full extension of the flight tracks above 8 km of altitude. By no means are these singular levels of hydrogen peroxide measured during a singular observation.

L257 (former L243): Slightly higher $H_2O_2$ levels were observed in the ITCZ (approx. 5°N–20°N), where locally mixing ratios up to 0.67 ppbv over a 1° x 1° bin of merged data were observed.

L263:  Good point.  If you flew at 15 km most of the time, the HOOH should be smaller.  What is your sampling profile?  The profile in Fig 3 is probably semi-uniform vertical sampling since it is limited to the LTO cycle above your airport.  What are the densities for the rest of the observations?

The general sampling profile is presented in Fig. 1 of the work, where the spatial extension of the flight tracks was presented. Please note the color coding according to the flight altitude. As stated in Sect. 2, the measurement flights focused on the upper troposphere. As discussed further in Sect. 3.5., due to the lack of statistically significant amount of data, the vertical profile information is restricted to 30 take-offs and landings at the base of operation at Sal. Please find below attached the latitudinal profiles for the entirety of the measurements and a general 3D overview of the performed flights with the respect to the measured $H_2O_2$ levels (Fig. 3 – 4).

[Figure]

Figure 3: Latitudinal dependence of hydrogen peroxide mixing ratios (mean ± 1σ) compared to EMAC simulations and calculations based on PSS (red: CEFE-Africa; black: PSS CAFE-Africa; blue: EMAC subdivided into three tropospheric regions: upper troposphere (a), free troposphere (b) and the boundary layer (c). The data with 6 min time resolution and mean values was binned for 2.5° of latitude. The corresponding numbers indicate the total amount of data points per bin. The shaded pattern marks the ITCZ region.

[Figure]

Figure 4: Overview of measurement flights with respect to the observed $H_2O_2$ levels (color-code) with the assumed region of ITCZ during the campaign highlighted in grey.

Generally, the data coverage density can be assumed from the temporal resolution of the measurement (1 data point per 6 min), the aircraft speed in the upper troposphere (being on average $232 \pm 13$ m/s) and the fact, that each measurement flight was performed once in the respective region. This gives a total of approx. 550 points and subsequently an average of approximately 40 data points per measurement flight excluding the area in proximity of the base of operation (16°35'–16°51'N; 22°52'–23°W). Further information on the data coverage density can be assumed from Fig. 5 of the work as well as Fig. S5 of the Supplement, where the numbers assigned in the plots indicate the total amount of data points with 6 min temporal resolution per 2.5° bin of latitude used to calculate the latitudinal profiles of the species.

L264: Agreed.

We thank the referee.

L266-269: This is an excellent way to express your 'latitude findings' rather what you have in the abstract. I am not sure they are dependent on latitude in the lower latitudes as tied to convection.

We thank the referee.

L290-291: I do not see this in Fig 3b at all. The for > 20N, the model:obs ratio varies from <0.3 to >3. How is this 'good'.

L306 (former L209 – 291) were changed according to the referee's comment.

L306 (former L290 – 291): Generally, the $H_2O_2$(EMAC)/$H_2O_2$(measurement) ratios indicate better agreement between the simulations and the measurements in the northern hemisphere ($\geq 20$°N; Fig. 3b). With decreasing latitude, the model tends to significantly overestimate hydrogen peroxide; $H_2O_2$(EMAC)/$H_2O_2$(measurement) ratios are increasing from approximately 2 to 4 with decreasing latitude.

L293: I do not believe the cloud scavenging bit – it is totally speculation. It could also be that EMAC has the wrong chemistry or photolysis. It might be scavenging, but really it could be anything.

As discussed further in the work in L379 – 390 (former L365 – 384) and in L426 - 428 (former L414 – 416) due to the rather minor differences between the measured and simulated $H_2O_2$ precursors ($HO_2$, OH and $j(H_2O_2)$), we assume the chemistry and photolysis not to be the major reason for the deviations between the model and the observations. From our past analyses we have learned that EMAC has difficulties to accurately simulate the cloud scavenging (Klippel et al. 2011; Hottmann et al. 2020; Hamryszczak et al. 2022). Thus, most likely, due to the enhanced presence of clouds in the region, as displayed in the Fig. 4 and Fig. S2 – S3, physical temporal as well as permanent loss of $H_2O_2$ due to scavenging may be responsible for the differences between the model simulations and measurements south from the ITCZ.
L309 (former L293) was changed.

L309 (former L293): Locally, most likely due to underestimated cloud scavenging as will be further discussed in this work, EMAC simulates highly elevated hydrogen peroxide with a factor of up to 14 higher than the measurements (4.5°N, 9.5°W).

L295: This is weird because one would have expected that Florence would have brought up lower HOH that was far from local PSS. Can you explain this?

As shown in Fig. 2, the tropical storm, Florence brought in the UT large concentration of $H_2O_2$. Nussbaumer et al. (2021), reported about increased levels of $H_2O_2$ in the convective outflow of the tropical storm. Additionally, at the same time, low NO mixing ratios were measured, indication an $O_3$ destructive regime, which might enhance the levels of peroxy radicals and subsequently the photochemical production of $H_2O_2$.

L303: Yes, whichever side of PSS (high or low) one would expect that recent covnection would NOT be in PSS. This is written incorrectly – the local PSS may be the only current P's and L's; but the amount deposited by convection is not in PSS. This section needs to be clarified.

L317 (former L303) was changed according to the referee's comment.

L317 (former L303): This indicates that, since the local photostationary steady-state conditions based on measured radical levels do not account for additional sources and sinks of $H_2O_2$, the observed discrepancy between the observations and local PSS are most likely related to transport and cloud scavenging.

L308-Fig4. I am not sure we get much from this that we did not see in Fig.3. It is close to the same information.

Other than in Fig. 3, where a ratio between the local PSS and the EMAC simulations versus the observations is presented, Fig. 4 summarizes the analysis results regarding the $H_2O_2$ production and loss dominated regions during the campaign. We compared the production term with the loss terms based on the measured $H_2O_2$ using the Eq. 2 & 3 presented in the work in Sect. 3.4. The shown $H_2O_2$ net production indicates clearly an imbalance between production and loss based on the observations and indicates a missing source/sink in the local budget of hydrogen peroxide.

Try 'sampled' instead of probed.

We thank the referee and follow his/her suggestion.

L27 (former L22): The significant influence of convection in the ITCZ and the enhanced presence of clouds towards the southern hemisphere indicate contributions of atmospheric transport and cloud scavenging in the sampled region.

L124 (former L113): The investigated area covered a latitudinal and longitudinal range from approximately 10°S–50°N and 50°W–15°E. The majority of vertical sampling was performed in close proximity of the base of operation and covered the altitudinal range between a few tens of meters above the surface and the maximal flown altitude (15 km). In sum, 30 takeoffs and landings with ascending and descending rates of 900 – 1100 m/min and 450 – 650 m/min, respectively, were performed giving an average descend/ascent rate of 775 m/min (with 1 point per 1550 m in vertical sampling at instruments temporal resolution of 2 min).

L151 (former L136): The hydroperoxide solution was sampled in two individual channels in response to addition of p-hydroxyphenyl acetic acid (POPHA) and horseradish peroxidase (HRP).

L240 (former L225): Vertical profiles of all species under investigation were calculated as 1000 m bins (means and medians) over the entire sampled atmospheric column.

L278 (former L262): This could be due to differences in the sampled altitudes, since ATom measurements were generally restricted to altitudes below 12 km.

L328 (former L312): Generally, the majority of the sampled region is loss-dominated, especially in the ITCZ (approx. 5°N–20°N) and towards the north, where an $H_2O_2$ deficit of up to approximately -0.01 ppbv h-1 was determined. $H_2O_2$ production-dominated regions of up to 0.03 ppbv h-1 are observed towards the south and in the coastal area.

L361 (former L345): The highest mean values of 0.49 (± 0.29) ppbv for PSS and 0.583 (± 0.40) ppbv for EMAC are found in the southernmost part of the sampled region.

L394 (former L382): Based on the ERA5 reanalysis results on cloud coverage during the measurement period (Hersbach, H. et al., 2019; Fig. S2), we hypothesize that the air masses sampled in this area were affected by cloud processing especially in the UT, causing the deviations towards the model calculations.

L431 (former L419): The absolute difference between the measured and calculated mixing ratios seems to be very prominent in and directly above the boundary layer (< 5 km) and can be associated with air masses affected by Saharan dust, which was often sampled during take-off and landings at Sal.

L479 (former L467): The local PSS calculations significantly underestimate the $H_2O_2$ mixing ratios in the north of the sampled region.

L482 (former L470): Further, the enhanced presence of clouds in the ITCZ and towards the southern hemisphere indicates significant cloud scavenging in the sampled region, justifying the deviations to the local photostationary steady-state calculations, which only account for photochemical sources and sinks of $H_2O_2$.

L348: This trend with latitude is meaningless, I am not sure how it is calculated but it does NOT look like the PSS curve in Fig.5. Also, what does it mean?

The trend was calculated based on a linear fit applied to the latitudinal average profile of all $H_2O_2$ data weighted by the standard deviation in the upper troposphere above 8 km of altitude between 10°S and 50 °N and states the general expected $H_2O_2$ mixing ratio trend with latitude based on local conditions. L364 (former L348) was changed according to the referee's comment.

L364 (former L348): Overall $H_2O_2$ mixing ratios from the PSS calculations show a decreasing tendency from the equator towards the subtropics.

I find this whole section up to L393 to be opaque, too much detail, and not convincing

L364 (former L348 – 393) where changed according to the referee's suggestion.

[revised manuscript text omitted]

L364: EMAC must be correct at all altitudes, particularly the convective source region. You cannot and do not address this issue. Did EMAC get the correct profile? why not compare with the Allen 2022 profiles of HOOH?

As displayed in Fig. 6 and discussed in L412 – 413 and L420 – 428 (former L400 – 401 and L408 – 416), EMAC simulates correct profiles. A comparison with former studies regarding the liability of the measurements is presented in Sect. 4.1.

L408: These are the sub-tropical profiles at 23N, not in the region where deep convection comes from. Where are the comparisons of OH and $HO_2$ profiles? Also, even at 23N EMAC shows the wrong profile with a peak ~ 1.5 km instead of 3 - 5km as observed. This is not great performance.

The presented vertical profiles were calculated based on the data at 16°35'–16°51'N, as stated in the caption of the Fig.6. For further clarification, please find the flight tracks of the entire campaign color-coded with respect to the observed $H_2O_2$ mixing ratios and the highlighted ITCZ region in Fig. 4 in this comment. As shown in Fig. S1 – 3 based on the monthly average data sets during the measurement period in the Supplement, the area of 16°35'–16°51'N is affected by deep convection.
The comparisons of the latitudinal and vertical profiles of the $H_2O_2$ precursors can be found in Fig. S5 and S6 in the Supplement of the work.
As discussed in L421 – 423 (former 409 – 411), due to the coarse model resolution of 1.8° x 1.8° a spatial restriction in resolving small-scale variations in hydrogen peroxide deposition processes in proximity to the base of operations on Cape Verde is presumably the main reason for the deviations in the vertical profiles.

L426-427: Fig.7 excess mole fraction of HOOH w.r.t. PSS does NOT tell me transport rates.

We apologize for the confusion. L438 (former L426 – 427) was changed according to the referee's comment.

L438 (former L426 – 427): Analogously, potential excess of $H_2O_2$ using model-simulated data was determined. The spatial distribution of the calculated excess $H_2O_2$ mixing ratios in the upper troposphere is presented in Fig. 7 as latitude vs. longitude plots of mean hydrogen peroxide levels aggregated over a spatial 1° x 1° grid at altitudes above 8 km.

L462: Correction – the ATom HOOH shows small latitudinal differences. However, the figure above is not published, so at least be specific as to which observations show a large latitudinal gradient.

We thank the referee for the correction. Unfortunately, we do not see the mentioned figure, but assuming the valuable information of the referee we changed our statement according to the comment.

L472 (former L462): However, the $H_2O_2$ mixing ratios measured during the CAFE-Africa campaign show only very little latitudinal variation over the Atlantic with a shift of the maximum mixing ratios towards the ITCZ.

Overall, this paper contains some reasonable conclusions, but spends too much time on weak points like the EMAC comparison.

We thank the referee for the comments and the assessment. Among the photochemical production and loss paths of $H_2O_2$, EMAC simulations picture sufficiently the general complexity of the chemistry and dynamics in the atmosphere taking also into account the influences arising from anthropogenic and natural emission sources. Thus, based on the comparison with EMAC simulations, we are not only able to link the deviations in observed $H_2O_2$ levels relative to the local PSS to non-photochemical processes, but also to indicate the main driving factors influencing the budget of hydrogen peroxide in the UT. For this reason, a comparison with the model output is of high importance to this work, as it highlights weaknesses of the model that must be considered in future analysis.

---

## Author Response (AR2)

**Measurement report: Hydrogen peroxide in the upper tropical troposphere over the Atlantic Ocean and western Africa during the CAFE-Africa aircraft campaign**

Zaneta Hamryszczak[1], Dirk Dienhart[1], Bettina Brendel[1], Roland Rohloff[1], Daniel Marno[1], Monica Martinez[1], Hartwig Harder[1], Andrea Pozzer[1,4], Birger Bohn[2], Martin Zöger[3], Jos Lelieveld[1,4], and Horst Fischer[1]

[1]Atmospheric Chemistry Department, Max Planck Institute for Chemistry, Mainz, 55128, Germany
[2]Institute of Energy and Climate Research, IEK-8: Troposphere, Forschungszentrum Jülich GmbH, Jülich, 52428, Germany
[3]Flight Experiments, German Aerospace Center (DLR), Oberpfaffenhofen, 82234 Germany
[4]Climate and Atmosphere Research Center, The Cyprus Institute, Nicosia, 1645, Cyprus

*Correspondence:* Zaneta Hamryszczak (z.hamryszczak@mpic.de) and Horst Fischer (horst.fischer@mpic.de)

**Point-by-point reply to the comment**

*Dear Dr Hamryszczak and co-authors,*

*I am happy to accept your manuscript for publication as measurement report in ACP after the following technical corrections:*

*- Please make sure that the title of your paper starts with "Measurement report:"*

*- Please make sure that the data you present in your paper can be accessed via a doi.*

*- Please make sure the the data are available under a non-restrictive license such as CC0 or CC BY.*

*Kind regards,*

*Gabriele Stiller*

Dear Dr Stiller,

Thank you for your comment.

The title of the paper was changed to "Measurement report: Hydrogen peroxide in the upper tropical troposphere over the Atlantic Ocean and western Africa during the CAFE-Africa aircraft campaign". Further, the data availability was changed according to the request and can be now accessed via a DOI under a non-restrictive license (https://doi.org/10.5281/zenodo.784589).

With best regards,

Zaneta Hamryszczak